# An *Inverse* Scaling Law for CLIP Training

**Xianhang Li**[*]    **Zeyu Wang**[*]    **Cihang Xie**

[*]equal contribution

UC Santa Cruz

## Abstract

CLIP, one of the pioneering foundation models that connect images and text, has enabled many recent breakthroughs in computer vision. However, its associated training cost is prohibitively high, imposing a significant barrier to its widespread exploration. In this paper, we present a surprising finding that there exists an *inverse* scaling law for CLIP training, whereby the larger the image/text encoders used, the shorter the sequence length of image/text tokens that can be applied in training. Moreover, we showcase that the strategy for reducing image/text token length plays a crucial role in determining the quality of this scaling law.

As a result of this finding, we are able to successfully train CLIP even with limited computational resources. For example, using **8** A100 GPUs, our CLIP models achieve zero-shot top-1 ImageNet-1k accuracies of **63.2%** in ~**2 days**. **67.8%** in ~**3 days**, and **69.3%** in ~**4 days**. Our method also works well when scaling up — with G/14, we register a new record of **83.0%** ImageNet-1k zero-shot accuracy, and meanwhile accelerate the training by ~**33×** compared to its OpenCLIP counterpart. By reducing the computation barrier associated with CLIP, we hope to inspire more research in this field, particularly from academics. Our code is available at https://github.com/UCSC-VLAA/CLIPA.

## 1   Introduction

Foundation models [56, 18, 43, 42] have emerged as a key driving force behind recent breakthroughs in multiple fields, including natural language processing [44, 6, 37, 12], computer vision [68, 46, 25, 50], and robotics [14], and have enabled groundbreaking real-world applications such as ChatGPT [38] and Stable Diffusion [49]. However, the development, training, and deployment of these models present significant challenges due to their high computational resource requirements and the need for specialized technical expertise, consequently restricting accessibility to a small group of researchers and technology companies.

We hereby focus on studying CLIP [42], one of the pioneering foundation models [25, 72, 27, 16, 51, 35] that bridge the gap between text and images and propels computer vision research into the "post-ImageNet" era. The impact of CLIP has been profound, not only in significantly advancing models' zero/few-shot capabilities and out-of-distribution generalization [42], but also in driving the development of the next generation of image-text foundation models, such as DALL-E [46] and Flamingo [2]. Although CLIP training is conceptually simple, reproducing CLIP has been challenging for researchers for years.

To increase the accessibility of CLIP, two significant milestones have been achieved: the OpenCLIP [24] project, which open-sourced the implementation of CLIP, and the release of LAION-400M and LAION-5B datasets [52], providing a wealth of high-quality image-text pairs for training. Yet, despite these strides, the cost of training associated with CLIP remains prohibitively high. For instance, replicating OpenCLIP-B/32's 62.9% zero-shot top-1 ImageNet-1k accuracy necessitates 36 hours of training with 128 A100 GPUs [24]. This cost is projected to rise considerably with the scaling law

37th Conference on Neural Information Processing Systems (NeurIPS 2023).

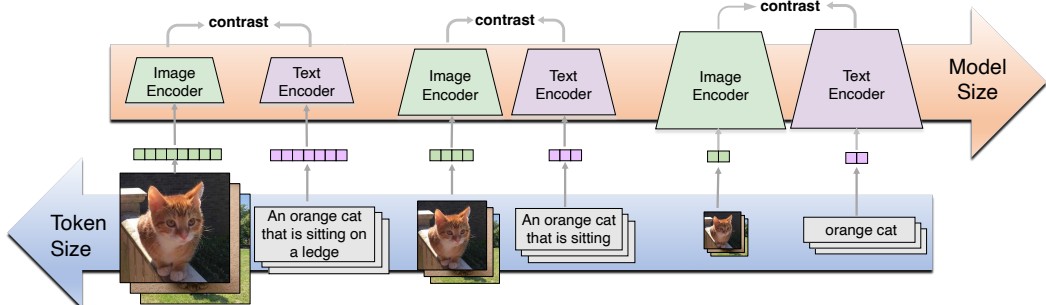

Figure 1: **The *inverse* scaling law for CLIP training.** It indicates that larger image/text encoders enable training with fewer image/text tokens while maintaining competitive performance.

[42, 11], which suggests that model performance typically scales proportionally with model size and the number of training tokens, thereby limiting the ability to explore CLIP more broadly and making it challenging for researchers to replicate and build upon these groundbreaking results.

In this paper, we report a surprising finding related to CLIP training that reveals an *inverse* scaling law. Specifically, we demonstrate that larger image/text encoders allow for the use of shorter image/text token sequences during CLIP training, with only a minor impact on performance. As illustrated in Fig. 1, while a small model S/16 requires a minimum image/text token length of 101/16 to avoid a noticeable performance drop (*e.g.*, within 1% in zero-shot ImageNet-1k [15] accuracy) compared to the vanilla training with the full token resolution, scaling up to L/16 can significantly reduce this requirement to a minimum image/text token length of 50/6. Additionally, it is worth noting that the strategy for reducing image/text tokens is critical, and those that maximize the retention of original (semantic) information tend to yield better scaling effects.

As a byproduct of this observation, we introduce CLIPA, a framework that can train CLIP efficiently and effectively at scale. For example, by training our CLIPA-L/16 for ~3 days on a server with eight A100 GPUs, it achieves a highly competitive **67.8%** zero-shot top-1 accuracy on ImageNet-1k. This performance stands in stark contrast to OpenCLIP-B/16, which attains a 67.1% zero-shot top-1 accuracy on ImageNet-1k but requires ~61 hours of training on 176 A100 GPUs [24], thereby costing over **16×** more GPU hours than our CLIPA-L/16. Our CLIPA can accelerate training more with bigger models — with G/14, CLIPA not only runs ~**33×** faster than OpenCLIP in training, but also impressively registers a record-high ImageNet-1k zero-shot top-1 accuracy of **83.0%.**

We hope this research will encourage a more diverse group of researchers, particularly those with limited computation resources, to delve into the exploration of CLIP training, or the training of foundation models in general.

## 2   Related Works

**Contrastive Language-Image Pre-training.** Over the past few years, the advent of CLIP [42] and ALIGN [25] has transformed the field of visual feature learning through language supervision. By exploiting vast quantities of web-scale data, these pioneering foundation models have shown exceptional zero-shot and out-of-distribution capabilities [42, 65, 70]. The streamlined design of CLIP has facilitated scaling to an unprecedented degree, resulting in substantial performance improvements. As a result, CLIP has been instrumental in empowering a wide range of applications, spanning from segmentation [64], video understanding [62], and image generation [40], to 3D understanding and manipulation [71, 57]. Furthermore, CLIP has played a vital role in catalyzing the development of next-generation image-text foundation models [46, 2, 49].

**Efficient CLIP Training.** The unparalleled success of CLIP hinges on the scale of both the data [42, 52, 25, 7, 66, 73] and the model [65, 37, 54]. While CLIP adheres impeccably to the scaling law [42, 11], it has also inadvertently sparked a race in large-scale training, one that is seemingly beyond the reach of many researchers in the field. This motivates the development of numerous efficient CLIP training methods. From the data perspective, de-replicating [41, 59, 1], re-sampling [61, 19, 31], and automated data curation [63] have been crucial in creating smaller but high-quality training datasets for accelerating training. On the other hand, FLIP [29] borrows the idea of masking

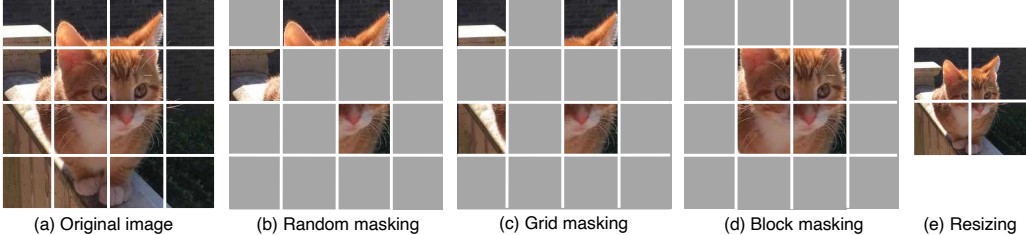

| (a) Original image | (b) Random masking | (c) Grid masking | (d) Block masking | (e) Resizing |

Figure 2: Visual comparison of different strategies for reducing image token length.

from Masked Image Modeling [21], and removes a large portion of the input image patches (50-75%) for fast CLIP training. The concurrent work RECLIP [28] shows resizing input images into a smaller size is a more effective strategy in speeding up training. Our work is based on FLIP, but it goes a step further by 1) exploring more effective semantic-preserving strategies for token length reduction in CLIP training; 2) pushing the idea of token length reduction to an extreme (*i.e.*, with only 17 images tokens and 8 text tokens), yielding a significant increase in training acceleration (up to $25\times$).

**Scaling law for Language Models.** The scaling law has emerged as a powerful tool, linking language model performance with model size, training data, and computational resources with a power-law relation [26]. This conclusion is empirically supported by the GPT model series [6, 37], T-5 [45, 13] and PaLM [12, 3] model families. In this paper, we focus on the scaling behavior of CLIP, but with two critical differences: 1) while the sample efficiency in the language model's scaling law is realized by using few training samples, we probe it by using fewer tokens in each image-text pair in CLIP training; 2) rather than comparing models of different sizes, our observation focuses on performance drop of the same model trained with input of various token lengths.

## 3 Reducing Image/Text Tokens

We study a total of eight token reduction strategies for CLIP training, four for image-based and four for text-based. Although many of these strategies have been extensively studied in the context of masked image/language modeling, such as random masking, which is generally the most effective, we observe that their effects on CLIP training are distinct.

### 3.1 Training Setup

Our training setup largely follows FLIP [29]. We use the vanilla ViT [18] as our visual encoder and the non-autoregressive Transformer [56] architecture as our text encoder. We train our models on the LAION-400M [52] dataset for 6.4 epochs, equivalent to ~2,000 ImageNet-1k epochs; this is then followed by a 0.36-epoch fine-tuning stage on full-resolution images ($224\times224$) with a maximum text length of 32. To ensure effective contrast between training samples, we set the batch size to $32k$. We apply a base learning rate of 8e-6 in the main training stage and 4e-7 in the fine-tuning stage. Gradient Checkpointing [8] is used to conserve GPU/TPU memory. Our data augmentation includes a simple random resizing crop with a minimum cropping ratio of 40%. Detailed hyperparameter settings and model configurations can be found in the appendix. We train L/16 CLIP models using various token reduction strategies and report the corresponding zero-shot top-1 accuracy on ImageNet-1k [15].

### 3.2 Image

We start our exploration with FLIP [29], which employs the random masking strategy from MAE [21] to reduce image token length during CLIP training. By setting the masking ratio to 75%, our re-implementation effectively reports a zero-shot top-1 ImageNet-1k accuracy of 67.6%.

In addition to random masking, we investigate two other strategies studied in MAE: *grid masking*, which preserves one patch in each $2 \times 2$ grid window, and *block masking*, which removes large blocks from the input. Fig. 2 provides visual examples of these three strategies at a 75% masking ratio. Intriguingly, while MAE deems random masking as the best strategy for masked image modeling, we notice that CLIP training has a differing preference. For example, grid masking attains a competitive zero-shot top-1 ImageNet-1k accuracy of 67.3%, while block masking is the most effective, achieving a zero-shot top-1 ImageNet-1k accuracy of 68.5%.

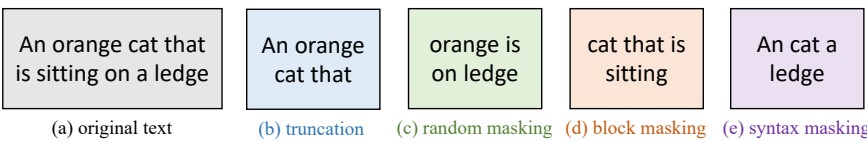

| (a) original text | (b) truncation | (c) random masking | (d) block masking | (e) syntax masking |

Figure 3: Visual comparison of different strategies for reducing text token length.

**Analysis.** We attribute this preference discrepancy to the two tasks' distinct learning natures. In masked image modeling, the objective is to generate absent information from a masked input. Therefore, strategies like random masking that effectively minimize retained information are preferred. In contrast, CLIP training aims to maximize information extraction from the input to achieve better discrimination between different samples. Strategies like block masking, which tend to preserve more structured patterns, can help models yield stronger performance.

**Resizing.** Building upon this analysis, we propose to apply image resizing as a more direct solution to retaining full image information. We use anti-aliasing bilinear interpolation as the resizing method to best preserve image quality. By training with the image resized to $112 \times 112$ (which is computationally equivalent to 75% masking), the L/16 model achieves a zero-shot top-1 ImageNet-1k accuracy of 68.9%. Notably, this simple resizing strategy surpasses all different mask strategies, highlighting the importance of retaining full input information in CLIP training.

### 3.3 Text

We next investigate how different strategies for reducing text tokens impact CLIP training. To speed up training, we default to resizing images to $112 \times 112$ as the image input. We begin with two techniques previously explored in FLIP: *truncation* and *random masking*. Truncation selects the first $N$ text tokens and discards the rest, while random masking randomly drops a portion of the text tokens. An illustrative example of these two strategies with a token length of 4 is shown in Fig. 3. By setting a maximum text token length of 8, truncation performs slightly better than random masking, resulting in a performance of 68.2% *vs.* 67.8%.

**Block masking.** We conjecture that the performance gain of truncation over random masking may be partially attributed to the use of consecutive text inputs. This leads us to investigate the efficacy of *block masking*, which randomly preserves consecutive text sequences during training. We limit the number of consecutive text tokens after masking to one for simplicity. With a maximum text token length of 8, this strategy achieves a competitive performance of 68.2%, outperforming random masking by 0.4%.

**Syntax masking.** Another potential approach to improving random masking is to assign different masking priorities to parts of speech. Specifically, we prioritize retaining nouns, followed by adjectives, and then other words. We refer to this strategy as syntax masking. With a maximum text token length of 8, syntax masking achieves the best performance among all strategies, recording a zero-shot top-1 ImageNet-1k accuracy of 69.0%.

In the next section, we systematically analyze how these four image-based strategies, namely, *random masking*, *grid masking*, *block masking*, and *image resizing*, and four text-based strategies, namely, *truncation*, *random masking*, *block masking*, and *syntax masking*, scale with varying token lengths across different model sizes.

## 4 An *Inverse* Scaling Law

**Training setup.** Models of three different scales are used: S/16, B/16, and L/16. Each model includes a visual encoder, namely ViT-S/16 (22M parameters), ViT-B/16 (87M parameters), and ViT-L/16 (304M parameters) [18]. In addition, we use text encoders with 33M, 53M, and 109M parameters, respectively. All these models are trained using the same setup outlined in Sec. 3.1, with one exception that a larger learning rate of 8e-7 is utilized during fine-tuning for S/16 and B/16.

### 4.1 Image

We first ablate how varying image token lengths affect CLIP training. Specifically, for random masking, block masking, and image resizing, we range the image token length from the full resolution

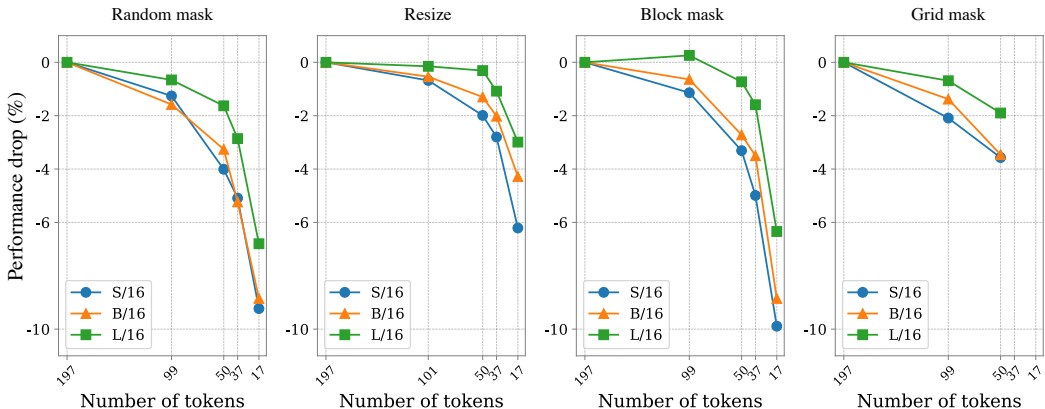

Figure 4: **The inverse scaling law on image tokens.** Compared to small models, larger models can utilize fewer image tokens to achieve the same performance drop to the full-resolution baseline.

(196 tokens) to an order of magnitude smaller one (16 tokens); for grid masking, the smallest length is set to 49 (i.e., selecting one in each $2 \times 2$ window), as it is non-trivial to further reduce it. Note we do not touch the setup for text encoders here, keeping the maximum length for text tokens as 32.

**Main Observation.** We analyze the zero-shot top-1 accuracy on ImageNet-1k [15] and plot the performance drop compared to the full resolution baseline in Fig. 4. Firstly, we note that performance generally decreases monotonically as token length reduces, which is expected given that models learn less information per sample. The only exceptional case occurs for block masking — when nearly halving the token length from 197 to 99, the performance for L/16 even slightly increases by 0.3%.

Furthermore, we observe that the performance drop for all four token reduction strategies becomes smaller as the model size increases. For instance, when reducing the token length from 197 to 17 using the resizing strategy, S/16 experiences a 6.2% performance drop, whereas scaling up the model size to B/16 reduces this drop to 4.3%; further using the considerably larger L/16 results in only a 3.0% performance drop. In other words, it suggests that larger models have the ability to achieve the same performance drop compared to the full-resolution baseline by utilizing fewer image tokens, as compared to their smaller counterparts. We term this phenomenon as the *inverse scaling law for CLIP training*, implying that by using larger models, we can train with fewer image tokens per sample while still delivering competitive performance.

Lastly, we find that the quality of this inverse scaling law strongly depends on how tokens are removed. More precisely, the more information that is retained, the smaller the length of tokens that can be applied during training. For instance, For instance, with a performance drop threshold of 2%, random masking requires 99 tokens for B/16 training. However, switching to image resizing, which retains substantially more image information, allows for a significant reduction in the minimum token length, down to 37.

For interested readers, we additionally offer two alternative views to understanding this scaling behavior in Fig. 9 (*i.e.*, model size *vs.* performance drop) and Fig. 10 (*i.e.*, token number *vs.* accuracy) in the Appendix.

**Zero-shot retrieval.** We further evaluate the image/text retrieval performance of CLIP with varying image token lengths on the challenging COCO [30] dataset. Fig. 5 shows the performance drop across different models for four image token reduction strategies. We note that, in most cases, the inverse scaling law proves consistent, as the degree of performance drop gradually decreases with increasing model size. For instance, using the random masking strategy that reduces the token length from 197 to 17, S/16 experiences a performance drop of 6.6% and 7.1% for image and text retrieval tasks, respectively. In comparison, the performance drops for B/16 are 5.8% and 5.9%, respectively; this performance drop is further reduced to 4.6% and 4.1% for L/16.

**Zero-shot robustness evaluation.** Fig. 6 reports robustness of the aforementioned models, tested on the ImageNet-V2 [47], ImageNet-R [22], ImageNet-A [23], and ImageNet-Sketch [58] datasets. We observe that, in most cases, larger models have a lesser performance drop than small models, which again confirms the validity of this inverse scaling law.

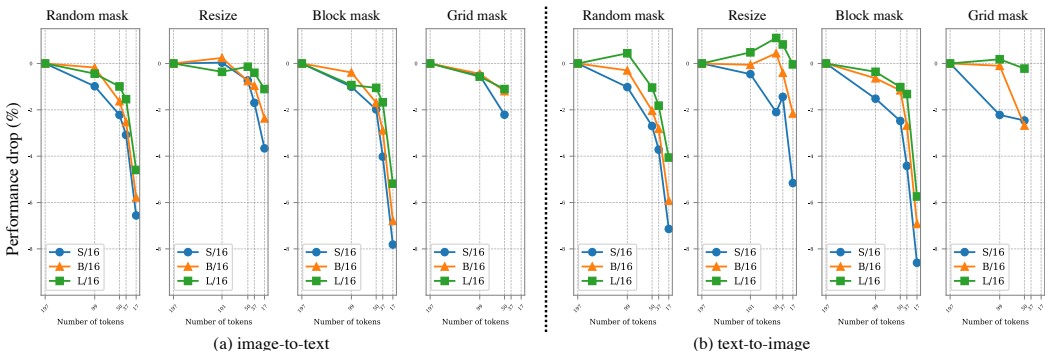

(a) image-to-text          (b) text-to-image

Figure 5: Zero-shot image/text retrieval performance on COCO [30]. Recall@1 is reported.

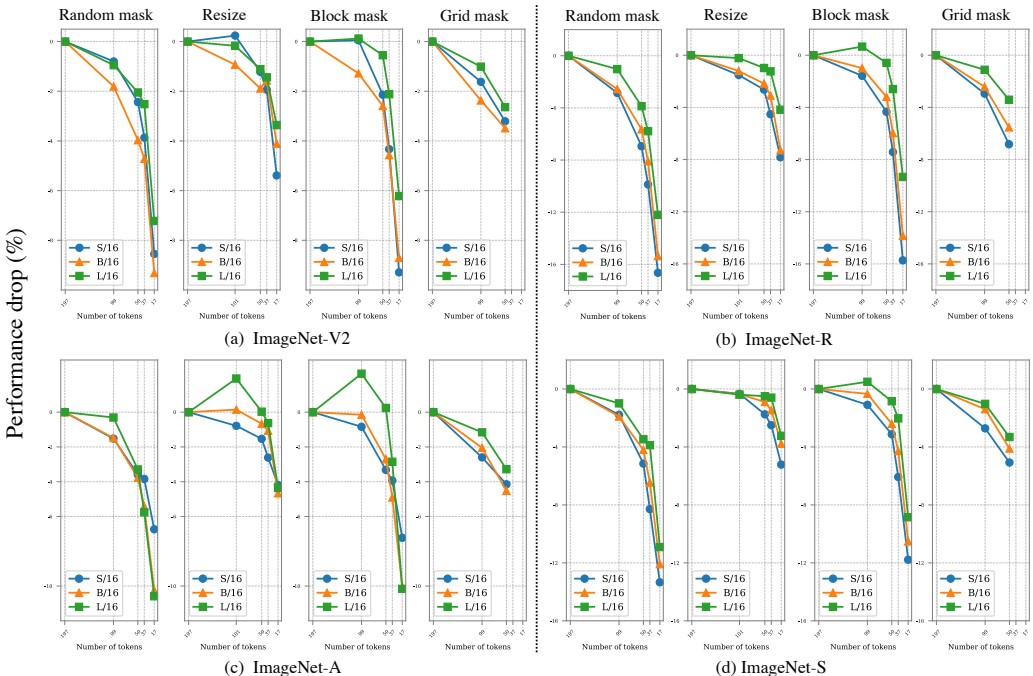

(a) ImageNet-V2          (b) ImageNet-R

(c) ImageNet-A          (d) ImageNet-S

Figure 6: Zero-shot robustness performance.

## 4.2 Text

We next study the impact of altering the maximum text token length on CLIP training. We use all four text reduction strategies introduced in Sec. 3.3, and for each strategy, we range the maximum text token length from 32 to 4. Additionally, to speed up training, we apply a resized $112 \times 112$ image as input, which runs $\sim 4\times$ faster than the $224 \times 224$ input, while only slightly affecting performance, *i.e.*, 0.3% drop on zero-shot top-1 ImageNet-1k accuracy for L/16.

**Main observation.** The data presented in Fig. 7 reflects a pattern similar to the one observed with image token, *i.e.*, the inverse scaling law is also evident when learning with text tokens. For example, when the maximum text length is set to 4 and the model size is scaled from S/16 to L/16, we observe a decrease in the performance drop from 5.7% to 5.2% for truncation, 3.4% to 2.0% for syntax masking, 4.3% to 2.9% for block masking, and 5.9% to 5.1% for random masking. Moreover, our analysis suggests that syntax masking is the most effective strategy for reducing text tokens, especially when setting the maximum text token lengths to be extremely short. For instance, with B/16 and a maximum text token length of 4, all other strategies incur a performance drop of more than 4.0%, whereas syntax masking results in a performance drop of merely 3.0%. Furthermore, we observe that for all strategies, the sensitivity of CLIP training to the reduction of text tokens remains relatively low until a threshold of 8 tokens is reached (*e.g.*, the performance drop is less than ~1.0%). However, beyond this point, the use of fewer text tokens leads to an abrupt performance drop.

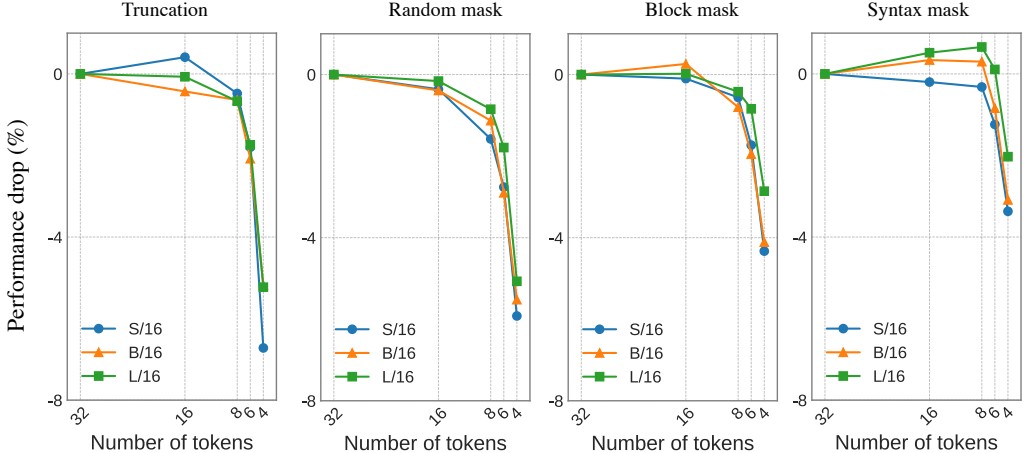

Figure 7: **The inverse scaling law on text tokens.** Similar to the observation with image tokens, larger models enable training with fewer text tokens while maintaining competitive performance.

Lastly, we notice another intriguing inverse scaling law uniquely related to syntax masking: reducing the text token length from 32 to 16 or 8 consistently enhances the performance of B/16 and L/16 models. This observation suggests that the language signal in our training data may be noisy, and filtering out certain information could potentially facilitate more effective representation learning.

**Zero-shot robustness & zero-shot retrieval evaluations.** We observe a similar trend for zero-shot robustness evaluation, where larger models typically yield smaller relative performance drops. In terms of zero-shot retrieval performance, for all four text token reduction strategies, we make two interesting observations: 1) there is almost no performance drop for all models when the text token length is reduced to 16; 2) further reducing the text token length to 8 or less, scaling up model size does not evidently help to reduce the performance drop. This second observation is expected, as reducing text length directly affects the capability to align image and text features in a fine-grained manner. Due to space limitations, we include the detailed results of the zero-shot image/text retrieval performance and the zero-shot robustness in Appendix.

### 4.3 ConvNeXt

In addition to ViT, we validate whether this inverse scaling law is also apparent within the context of CNN architectures. For this analysis, we select ConvNeXt [32], given its outstanding performance on various visual benchmarks. Although different masking strategies are applicable for ConvNeXt, they can only offer a modest training speedup due to the lack of computationally efficient support for sparse convolution [60]. However, image resizing emerges as a viable strategy for expediting CLIP training with ConvNeXt, as it avoids the need of using sparse convolution [55, 20].

We focus on studying ConvNeXt-T and ConvNeXt-B, which are of a similar scale to ViT-S/16 and ViT-B/16, respectively. We utilize the same training setup as for ViT, and incorporate additional augmentations [9, 10]. The full results are listed in Appendix.

**Main Observation.** We observe that ConvNeXt-B consistently shows a smaller performance drop than ConvNeXt-T when a smaller input size is applied. By setting a performance drop of 1.5% as the threshold, we find that while ConvNeXt-T necessitates an input image size of $112 \times 112$, scaling to ConvNeXt-B enables further reduction of the input size to $96 \times 96$. These observations confirm the existence of the inverse scaling law for ConvNeXt in CLIP training.

## 5 Training CLIP with Limited Resources

Our discussions in Sec. 4 reveal that larger models have the ability to train with fewer tokens while still preserving competitive performance. This ability brings substantial practical benefits, including improved memory footprint and faster training speed. In this section, we showcase how this inverse scaling law can be leveraged to train CLIP models efficiently and effectively, particularly when computational resources are limited.

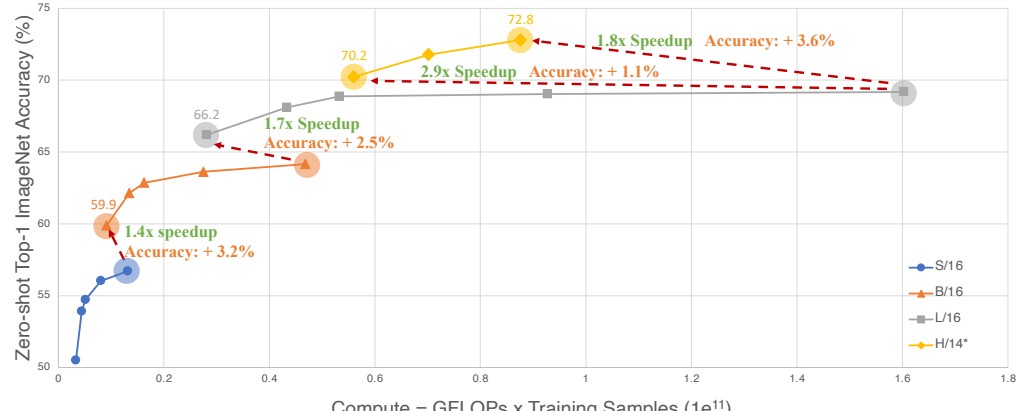

Figure 8: **Accuracy *vs.* compute trade-off.** The x-axis shows overall training cost, and the y-axis shows corresponding ImageNet-1k zero-shot accuracy. The models are trained with different token lengths, resulting in varying compute costs. *\* indicates the application of additional color jitter and grayscale augmentation, as well as the use of global average pooling instead of the classification token. These modifications are found to be beneficial for stabilizing training with reduced token lengths in large models.*

| model | samples@image resolution | GPU hours | zero-shot classification | | | | | | zero-shot retrieval | | | |
| | | | | | | | | | COCO | | Flickr30k | |
| | | | IN-1k | IN-V2 | IN-A | IN-R | ObjectNet | IN-Sketch | image | text | image | text |
|---|---|---|---|---|---|---|---|---|---|---|---|---|
| OpenAI-B/32, Our Eval | 12.8B@$224^2$ | 4600 | 63.3 | 55.9 | 31.6 | 69.3 | 44.2 | 42.3 | 30.4 | 50.2 | 58.9 | 77.6 |
| OpenAI-B/16, Our Eval | 12.8B@$224^2$ | 10700 | 68.3 | 61.9 | 49.9 | 77.7 | 55.3 | 48.2 | 33.1 | 52.4 | 62.1 | 81.9 |
| OpenAI-L/14, Our Eval | 12.8B@$224^2$ | 50800 | 75.5 | 69.8 | 70.8 | 87.8 | 68.9 | 59.6 | 36.5 | 56.4 | 65.3 | 85.1 |
| OpenCLIP-B/32, Our Eval | 12.8B@$224^2$ | 4600 | 62.9 | 55.1 | 21.7 | 73.4 | 48.9 | 49.4 | 35.3 | 52.6 | 61.7 | 79.0 |
| OpenCLIP-B/16, Our Eval | 12.8B@$224^2$ | 10700 | 67.1 | 59.6 | 33.2 | 77.9 | 51.5 | 52.4 | 38.3 | 55.4 | 65.5 | 83.3 |
| OpenCLIP-L/14, Our Eval | 12.8B@$224^2$ | 50800 | 72.8 | 65.4 | 46.5 | 84.9 | 59.9 | 59.6 | 43.0 | 59.7 | 70.3 | 87.6 |
| CLIPA-B/16 (I50,T16) | 2.56B@$112^2$+128M@$224^2$ | 444 | 63.2 | 55.6 | 26.8 | 73.2 | 44.3 | 48.7 | 35.2 | 53.1 | 58.3 | 75.3 |
| CLIPA-L/16 (I17,T16) | 2.56B@$64^2$+128M@$224^2$ | 628 | 67.8 | 60.4 | 38.3 | 81.2 | 52.8 | 56.4 | 40.1 | 58.4 | 64.0 | 81.5 |
| CLIPA-L16 (I37,T8) | 2.56B@$96^2$+128M@$224^2$ | 826 | 69.3 | 61.7 | 43.6 | 84.0 | 55.4 | 58.7 | 39.8 | 56.8 | 67.5 | 81.9 |

Table 1: **Training CLIPA with limited resources.** CLIPA models are first pre-trained with smaller token lengths with 2.56B training samples and subsequently fine-tuned with full token lengths with 128M epochs on LAION-400M. These models are trained on an 8-A100 GPU machine. '(I$X$,T$Y$)' indicates the model is pre-trained with an image token length of $X$, and a maximum text token length of $Y$. Image resizing and text truncation are used for token length reduction.

We start by recasting the image resizing results of Fig. 4 in the context of *computation vs. performance* shown in Fig. 8. In addition to the clear performance advantage of larger models over smaller ones, an interesting observation is that this inverse scaling law offers the potential for faster and more powerful CLIP training. For instance, our L/16 model, using a total image token length of 17, outperforms the standard B/16 model setup (*i.e.*, with a total image token length of 197) by 2.5%, while achieving a 1.7× speedup. This process can be further accelerated by training with fewer text tokens, especially when employing a large text encoder (*e.g.*, as in H/14).

Motivated by the above observations, we introduce an effective and efficient CLIP training strategy: training with a larger model but with reduced input token lengths. This approach, dubbed as *CLIPA*, enables CLIP training even with academic resources. The training setup of CLIPA follows the protocol outlined in Section 3.1, with the addition of color jitter and grayscale image augmentation [9, 10], and the usage of global average pooling in ViT [29, 21]. To reduce the token length in CLIP training, image resizing and text truncation are used by default. More training details can be found in Appendix. All these models are trained using the OpenCLIP codebase [24] in PyTorch [39] on a machine equipped with 8 NVIDIA A100 GPUs.

As demonstrated in Tab. 1, our CLIPA provides both faster training times and improved performance in comparison to OpenCLIP. For example, our CLIPA-B/16 surpasses the vanilla OpenCLIP-B/32 baseline by 0.3% on zero-shot ImageNet-1k classification, more importantly, requiring ∼**10**× fewer GPU hours. Similarly, our CLIPA-L/16 outperforms the vanilla OpenCLIP-B/16 baseline by 0.7%, yet consumes **17**× fewer GPU hours. Notably, our CLIPA-B/16 can be trained on an 8-A100 GPU machine in ∼2 days, and CLIPA-L/16 in ∼3 days, highlighting the efficiency and effectiveness of CLIPA in facilitating CLIP training while preserving competitive performance.

| model | data source | samples@image resolution | compute(1e12) | zero-shot classification | | | | | | zero-shot retrieval | | | |
| | | | | IN-1k | IN-V2 | IN-A | IN-R | ObjectNet | IN-Sketch | COCO image | COCO text | Flickr30k image | Flickr30k text |
|---|---|---|---|---|---|---|---|---|---|---|---|---|---|
| FLIP-H/14, Our Eval | LAION-2B | 25.6B@224² + 128M@224² | 2.4 | 78.4 | 71.7 | 60.3 | 90.8 | 69.4 | 67.5 | 49.9 | 67.0 | 78.3 | 93.4 |
| OpenCLIP-H/14 | LAION-2B | 32B@224² | 5.7 | 78.0 | 70.8 | 59.2 | 89.3 | 69.7 | 66.6 | 49.5 | 66.0 | 77.8 | 90.8 |
| OpenCLIP-G/14 | LAION-2B | 32B@224² + 6.7B@224² | 29.8 | 80.1 | 73.6 | 69.4 | 92.2 | 73.0 | 68.9 | 51.4 | 67.3 | 79.6 | 92.9 |
| CLIPA-H/14 (I36,T8) | LAION-2B | 12.8B@84² + 128M@224² | 0.4 | 77.9 | 71.4 | 66.2 | 91.3 | 71.1 | 68.4 | 49.3 | 66.9 | 77.2 | 91.0 |
| | | 12.8B@84² + 512M@224² + 128M@336² | 0.4 | 79.1 | 72.3 | 71.7 | 92.7 | 69.9 | 70.0 | 50.2 | 67.5 | 78.2 | 92.3 |
| CLIPA-H/14 (I36,T8) | DataComp-1B | 12.8B@84² + 128M@224² | 0.4 | 81.5 | 75.0 | 76.9 | 94.3 | 74.1 | 72.7 | 49.1 | 67.0 | 75.7 | 90.6 |
| | | 12.8B@84² + 512M@224² + 128M@336² | 0.4 | 81.8 | 75.6 | 82.7 | 94.4 | 77.4 | 72.8 | 49.2 | 67.2 | 76.3 | 90.3 |
| CLIPA-G/14 (I36,T8) | DataComp-1B | 12.8B@84² + 512M@224² | 0.8 | 82.7 | 76.9 | 81.7 | 95.1 | 77.1 | 74.3 | 50.0 | **67.9** | 77.7 | 91.8 |
| | | 12.8B@84² + 512M@224² + 128M@336² | 0.9 | **83.0** | **77.3** | **85.9** | **95.4** | **79.7** | **74.5** | 50.4 | 67.8 | 78.2 | 92.1 |

Table 2: **Training CLIPA at scale.** CLIPA models are first pre-trained with smaller token lengths with 12.8B pre-training samples and subsequently fine-tuned with full token lengths. The Compute cost is measured in the GFLOPs of the model times the number of samples seen during training. '$(IX,TY)$' indicates the model is pre-trained with an image token length of $X$, and a maximum text token length of $Y$.

**H/14 model.** We hereby include a bigger model, CLIPA-H/14, for experiments. Note that, here we cut the input text token length from 32 to 8, yielding an additional ∼1.3× training speedup. These results are added to Fig. 8. With an input image size of 84 and a text token length of 8, our CLIPA-H/14 achieves a compelling zero-shot top-1 ImageNet-1k accuracy of 72.8%. This performance is on par with that of OpenCLIP-L/14, while the total computational requirement is reduced by ∼**25**×.

## 6 CLIPA at Scale

In this section, we delve deeper into the scaling behavior of CLIPA with larger models (*e.g.*, G/14) and larger datasets (*i.e.*, LAION-2B [52] and DataComp-1B [19]). We default to the setup of 12.8B pre-training samples. We find that extending the fine-tuning schedule at a resolution of $224 \times 224$ from 128M to 512M training samples, followed by another 128M samples's training at $336 \times 336$ resolution, demonstrates a clear improvement with the H/14 model (79.1% *vs*. 77.7%). Moreover, our updated fine-tuning schedule incorporates a random masking strategy at both resolutions (30% for $224 \times 224$ and 40% for $336 \times 336$), which reduces the training overheads by a large margin with little-to-no performance drop. More details can be found in the Appendix.

**Main results.** As shown in Tab. 2, when trained on the same dataset LAION-2B and with the $224 \times 224$ resolution, our CLIPA-H/14 attains comparable performance with OpenCLIP-H/14 but merely with ∼**1/15** training computations. This signifies a remarkable decrease in cost – *e.g.*, given that the training cost for the reported OpenCLIP result amounts to ∼5,600 GPU-days, CLIPA could save ∼5,230 GPU-days. Additionally, compared with FLIP-H/14, our CLIPA-H/14 achieves a better 79.1% ImageNet-1k performance but can be 6× faster.

When continuing scaling our model size to G/14, with the same number of seen samples from the DataComp-1B [19] dataset, we successfully establish a new state-of-the-art open-sourced ImageNet-1k zero-shot accuracy of **83.0%**. Notably, this is achieved with ∼**33** × less computation compared with previous OpenCLIP-G/14 model. These findings could potentially pave the way for the training of even larger models on even larger datasets, particularly for those with substantial access to GPU/TPU resources.

To further evaluate the performance of our approach, we also evaluate our CLIPA-H/14 model on the VTAB benchmark [69]. The results are included in Tab. 3. On this highly diverse and challenging set of vision tasks, CLIPA still achieves comparable or even superior performances but with significantly less training cost, demonstrating its good generalizability.

## 7 Limitation

The recent work [67] shows that CLIP models generally are limited at capturing relationships, attributes, and order information. To give a more comprehensive evaluation, we compare our CLIPA model with OpenCLIP on the ARO benchmark [67], a dataset created to evaluate the ability to understand different types of relationships, attributes, and order information. The results are shown in the Appendix (Tab. 14). We can observe that, while OpenCLIP-B/16 slightly outperforms CLIPA-B/16, the absolute performance of both models remains somewhat limited.

| model | samples | compute | IN-1k | Caltech101 | Cifar10 | Cifar100 | Clevr_all | Clevr_dis | Dmlab | Dtd | Eurosat | Kitti | Flowers | Pets | Pcam | Resisc45 | Smallnorb_A | Smallnorb_E | Svhn |
|---|---|---|---|---|---|---|---|---|---|---|---|---|---|---|---|---|---|---|---|
| OpenCLIP-H/14 | 32B | 5.7 | 78.0 | **84.9** | 97.5 | 84.7 | **26.8** | 23.6 | 14.0 | 67.8 | **72.6** | 11.0 | 80.1 | 94.5 | 54.2 | 69.9 | 5.4 | 11.1 | 56.3 |
| FLIP-H/14, Our Eval | 25.6B | 2.4 | 78.4 | 84.3 | **98.2** | 86.9 | 16.8 | **24.7** | **19.7** | 67.5 | 64.5 | **16.5** | **81.1** | 95.3 | 48.5 | **70.8** | 5.4 | 11.0 | 48.6 |
| CLIPA-H/14 (I36,T8) | 12.8B+512M | 0.4 | 77.9 | 84.8 | 98.1 | **87.4** | 21.2 | 24.3 | 15.5 | 70.2 | 66.8 | 16.0 | 77.9 | 93.8 | 56.0 | 69.7 | 5.7 | 11.6 | 53.4 |
| CLIPA-H/14 (I36,T8) | 12.8B+512M+128M | 0.4 | **79.1** | 84.6 | **98.2** | 86.4 | 17.5 | 21.6 | 14.0 | **71.4** | 64.0 | 15.8 | 79.6 | 94.5 | **58.1** | 70.3 | **5.8** | **13.9** | **59.6** |

Table 3: **Comparison on VTAB benchmarks** by zero-shot top-1 accuracy. All models are trained on the LAION-2B dataset. Entries in **bold** are best results. Compute is measured in GFLOPs (1e12).

To mitigate this relational understanding issue, a composition-aware hard negative mining strategy (NegCLIP) is introduced in [67]. Note that this strategy is extremely lightweight, and can be seamlessly integrated as an additional fine-tuning stage in enhancing CLIP's text understanding ability. Our results in Tab. 14 also corroborate the efficacy of NegCLIP, e.g., both OpenCLIP and CLIPA nearly double their performance on benchmarks like COCO-Order and Flickr30k-Order. Concerning the initial underperformance on ARO benchmarks, we leave it as a future work.

# 8   Conclusion

In this paper, we delve deep into CLIP training. Our investigation unveils an intriguing inverse scaling law, suggesting that larger models require fewer input tokens during training. Moreover, among the eight token reduction strategies we studied, we identify that resizing for image input and syntax masking for text input provides the best overall scaling quality. This finding underscores the crucial role of semantic information preservation in efficient CLIP training. Our findings can enable significantly faster CLIP training with better results, especially given limited resources. We hope that our work could encourage a wider range of researchers, particularly those lacking access to substantial computational resources, to engage more in exploring CLIP training.

# 9   Broader Impact

Large foundation models trained by language supervision have emerged as a pivotal force driving recent advancements in the language and vision domain. Our discovery of the inverse scaling law has democratized access to this technology, enabling the training of proficient CLIP models on a modest budget. This breakthrough has substantial environmental implications, as it significantly curtails tens of thousands of GPU/TPU hours, thereby reducing energy consumption and associated carbon emissions. It is also worth mentioning that our models are trained on publicly available web-scale datasets [53, 52]. Therefore, the derived weights may inadvertently mirror any bias or harmful contents inherent in the training sets. As such, care should be taken when interpreting the outputs of such models and deploy them in the real-world applications.

# Acknowledgement

This work is supported by a gift from Open Philanthropy, TPU Research Cloud (TRC) program, and Google Cloud Research Credits program.

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

# A  Implementation Details

| model | Embed dim | Vision Transformer | | | Text Transformer | | | # params (M) | | |
|---|---|---|---|---|---|---|---|---|---|---|
| | | layers | width | heads | layers | width | heads | vision | text | total |
| S/16 | 384 | 12 | 384 | 6 | 12 | 384 | 6 | 22 | 33 | 55 |
| B/16 | 512 | 12 | 768 | 12 | 12 | 512 | 8 | 86 | 53 | 141 |
| L/16 | 768 | 24 | 1024 | 16 | 12 | 768 | 12 | 303 | 109 | 414 |
| H/14 | 1024 | 32 | 1280 | 16 | 24 | 1024 | 16 | 631 | 334 | 967 |
| G/14 | 1280 | 48 | 1664 | 16 | 32 | 1280 | 20 | 1844 | 672 | 2516 |

Table 4: **CLIP [42] model configurations** used in our paper.

| Config | Value |
|---|---|
| optimizer | AdamW [34] |
| optimizer momentum | (0.9, 0.95) |
| batch size | 32768 |
| base lr | 8e-6 |
| minimal lr | 0 |
| warm-up steps | 1600 |
| schedule | cosine decay [33] |
| weight decay | 0.2 |
| random crop area | (40, 100) |
| resize method | bi-linear |
| color jitter [9] | 0.32 |
| temperature init | 1/0.07 [24, 29] |

Table 5: **Pre-training hyper-parameters**

## A.1  Architectures

Our experimental results are based on specific model configurations shown in Tab. 4, following FLIP [29]. Our visual encoder architecture employs three different scales (S/16, B/16, and L/16) with the same patch size, allowing us to investigate the effect of scaling. In our CLIPA models, we employ vanilla ViT [18] with global average pooling as the visual encoder. The sine-cosine positional embeddings are used in ViT [56]. As for text encoder, we adopt the non-autoregressive Transformer [56, 29] and employ a WordPiece tokenizer [17], which includes a "CLS" token for each input text. To ensure uniformity in the input length, we apply zero-padding to those input texts that are shorter than the maximum token length of our model. For the ConvNeXt [32], we employ the same mode configuration as described in [32], and follow the setting in [24].

## A.2  Hyper-parameters

**Pre-training.** Our CLIPA pre-training configuration is outlined in Tab. 5. Notably, due to limited resources, we use a base learning rate of 8e-6 and a smaller $32k$ batch size. In addition, we apply a color jitter augmentation of strength 0.32 and probability 0.8, and a gray-scale augmentation of probability 0.2 [36, 9, 10].

**Fine-tuning.** Following pre-training, we conduct a short-period fine-tuning of the models using full-resolution images of size $224 \times 224$ and texts with a maximal length of 32. The fine-tuning process consists of 4000 steps with an 800-step warm-up period. The base learning rate for fine-tuning is set to 8e-7, while all other parameters remain the same as those used during pre-training. Note that due to limited computation resources, our CLIPA-B/16 and CLIPA-L/16 are fine-tuned with $8k$ and $7k$ batch size. The results of different fine-tuning batch size are shown in Tab. 6. It can be observed that a batch size of $8k$ already achieves competitive performance compared to a batch size of $32k$.

**Implementation.** We implement two codebases based on JAX [5] and Pytorch [39] respectively. Our JAX codebase is built on Big Vision [4] and our pytorch code base mainly followed OpenCLIP [24]. Most of our experiments are conducted with TPU-V3, except that CLIPA-B/16 and CLIPA-L/16 in Tab. 1 are conducted with A100 GPUs.

**CLIPA-H/14 and G/14.** For comparison with previous state-of-the-art models in the scaling experiments, we follow the approach in FLIP [29], using a 64k batch size for pre-training, and adjusting the warmup steps to 3200 to mitigate the unstable training of larger models. For fine-tuning at a 224 resolution, we apply random masking for image encoder with a 30% mask ratio to expedite the

| Model | 32k | 16k | 8k | 4k | 2k |
|---|---|---|---|---|---|
| CLIPA-L/16 ((I17,T16) | 68.1 | 67.8 | 67.7 | 67.1 | 66.8 |

Table 6: Ablation on fine-tuning batch size.

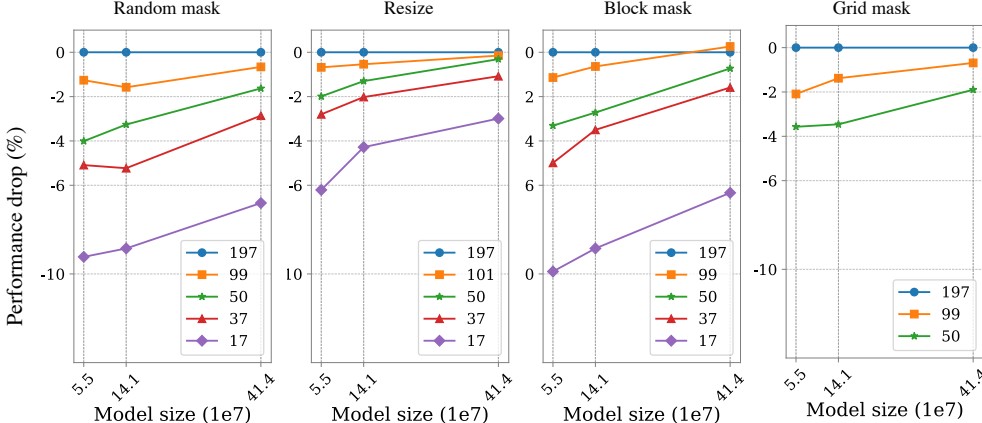

Figure 9: **Model size *vs*. Performance drop.** Different lines are for different total numbers of input image tokens per sample.

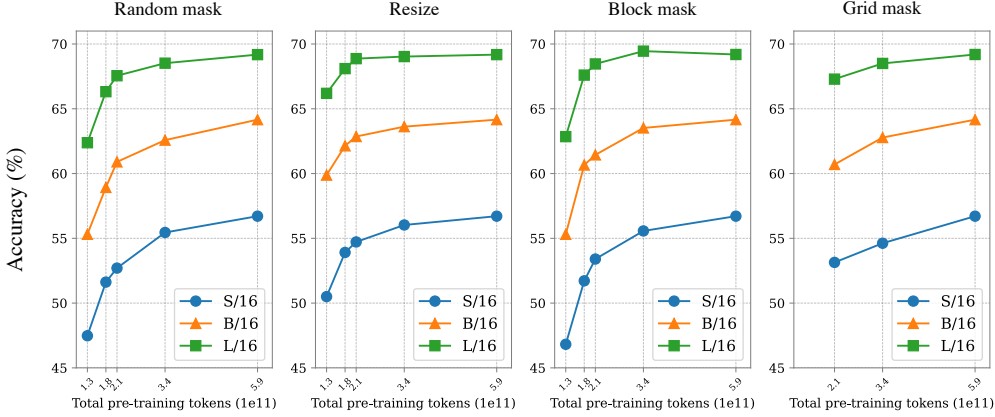

Figure 10: **Total number of pre-training tokens *vs*. Accuracy.**

training process and reduce memory costs. And we utilize a 32k batch size with a learning rate of 4e-7 and train with 512M training samples. In fine-tuning at a 336 resolution, the mask ratio is set at 40% for both models. We further reduce the base learning rate to 1e-7 and train an additional 128M samples with a batch size of 16k. These two models are trained on a 256-core TPU-V3 pod. We incorporate a model-sharding strategy in our G/14 model, which is based on the T5X implementation [48]. Apart from this, we employ a distributed data-parallel strategy.

## A.3 Evaluation Setting

Our evaluation protocol is largely based on the original CLIP paper [42], and we employ the benchmarking tool provided by OpenCLIP [24]. To evaluate our model on ImageNet-1k [15], we use 80 prompt templates for zero-shot testing. Following OpenCLIP [24], we resize the shorter side of the input images to 256, and then perform a center crop of size $224 \times 224$. When a larger resolution of 336 is adopted, we directly resize the image into $336 \times 336$ without cropping.

| model | GAP | color-jitter [9] | pytorch-impl. | fine-tuning batch size | pre-train | fine-tune |
|---|---|---|---|---|---|---|
| Baseline | ✗ | ✗ | ✗ | 32k | 58.3 | 66.3 |
| | ✔ | ✗ | ✗ | 32k | 59.8 | 67.8 (+1.5) |
| | ✗ | ✔ | ✗ | 32k | 58.5 | 67.1 (+0.8) |
| | ✔ | ✔ | ✗ | 32k | 59.9 | 68.1 (+1.8) |
| | ✔ | ✔ | ✗ | **7k** | 59.9 | 67.6 (+1.3) |
| | ✔ | ✔ | ✔ | **7k** | 60.3 | 67.8 (+1.5) |

Table 7: **Training details analysis.** We use CLIPA-L/16 (I17, T16) model as baseline and report ImageNet-1k [15] zero-shot top-1 performance. GAP: global average pooling in visual encoder. Pytorch-impl. : we re-implement JAX [5] version and reproduce the results with Pytorch [39] on GPUs

# B  More Results

In this section, we present more detailed results, including an alternative view of inverse scaling law, detailed ablation studies of training details and fine-tuning, and numeric results on ImageNet-1k, which we used to plot Fig. 4 in the main text.

## B.1  Alternative view of inverse scaling law

For improved representation, we offer two alternative interpretations of Fig. 4 in Fig. 9 and 10. In Fig. 9, we plot the model size on the x-axis, depicting fractions of token reduction as separate lines. In Fig. 10, the total pre-training tokens are used on the x-axis. In both perspectives, it's apparent that larger models demonstrate a significantly smaller performance decrease when using fewer tokens for pre-training.

| model | # image token | # text token | data source | # seen samples | total compute ($\times 1e11$) | IN-1k |
|---|---|---|---|---|---|---|
| CLIPA-L/16 | 36 | 8 | LAION-400M | 2.56B + 128M | 0.5 | 69.3 |
| | | | LAION-400M | 2.56B + 128M | 0.8 | 72.8 |
| CLIPA-H/14 | 36 | 8 | **LAION-2B** | 2.56B + 128M | 0.8 | 74.1 |
| | | | LAION-2B | **12.8B + 128M** | 4 | **77.9** |

Table 8: **Scaling up CLIPA.** Specifically, we explore scaling from the aspects of data, model, and schedule. We pretrain the H/14 model with 36 image tokens ($84 \times 84$) and 8 text tokens; for fine-tuning, we use 256 ($224 \times 224$) image tokens and 32 text tokens.

## B.2  Ablation

**Ablation on Training details**  We present a comprehensive analysis of the training details employed to CLIPA. The results are summarized in Tab. 7. First, using global average pooling in ViT, instead of class token as in [24], leads to a substantial improvement of ~1.5% in ImageNet-1k zero-shot accuracy. Second, we also observe that incorporating stronger augmentation techniques [9] leads to a ~0.8% improvement. Together, they yield a notable 1.8% improvement. It is also worth mentioning that to ensure a fair comparison, we switch to the widely-used OpenCLIP codebase [24], which is implemented in PyTorch [39]. Finally, to accommodate the limited GPU memory, we employ a batch size of 7k for fine-tuning the CLIPA-L/16 model. Our experiments demonstrate that this adjustment results in only a marginal decrease in performance.

**Ablation on scaling up.**  We next investigate the scaling behavior of CLIPA. Specifically, our scaling efforts cover three aspects: model, data, and training schedule. The results are reported in Table 8.

First, we can observe that scaling the model size from L/16 to H/14 boosts the performance from 69.3% to 72.8%. Furthermore, we note switching the training dataset from LAION-400M [53] to LAION-2B [52] yields another 1.3% improvement, suggesting the importance of data diversity. Lastly, by increasing the training schedule by a factor of 5, resulting in a total of ~13B seen samples, we achieve an impressive performance of 77.9%. We stress that this scaled version of CLIPA-H/14 model readily outperforms its counterpart in FLIP [29] by 0.3% while requiring only $1/3$ of the training budget.

These results confirm the efficiency and effectiveness of training CLIPA at scale. Next, we set this CLIPA-H/14 with 77.9% performance as our baseline for further ablation in the fine-tuning stage.

| masking ratio | random | block | grid |
|---|---|---|---|
| 0% | 77.9 | 77.9 | 77.9 |
| 25% | **78.2** | 78.0 | 77.9 |
| 50% | **77.7** | 77.6 | 77.6 |
| 75% | **76.2** | 74.3 | **76.2** |

Table 9: Comparison of different masking strategies. The results are obtained on on the LAION-2B dataset with H/14 model.

| case | masking ratio | resolution | # seen samples | training FLOPs | IN-1k |
|---|---|---|---|---|---|
| baseline | 0% | $224^2$ | 128M | 177.0G | 77.9 |
| (1) | 30% | $224^2$ | 128M | 135.9G | 78.0 |
| (2) | 30% | $224^2$ | 512M | 135.9G | 78.6 |
| (3) | 30% | $224^2$ | 640M | 135.9G | 78.5 |
| (4) | 40% | $336^2$ | $640M$ | 237.8G | 78.9 |
| (5) | 30%+40% | $224^2 + 336^2$ | 512M+128M | 156.3G | **79.1** |

Table 10: **Ablation on fine-tuning schedule and masking.** In case (5), we use $224 \times 224$ input with a masking ratio of 30% for the first 512M samples, and $336 \times 336$ input with a masking ratio of 40% for the rest 128M samples.

| Masking strategy | Masking ratio | # of tokens | S/16 | | B/16 | | L/16 | |
|---|---|---|---|---|---|---|---|---|
| | | | pre-train | fine-tune | pre-train | fine-tune | pre-train | fine-tune |
| baseline | 0.0% | 197 | - | 56.7 | - | 64.2 | - | 69.2 |
| random | 50.0% | 99 | 54.7 | 55.5 | 61.9 | 62.6 | 68.3 | 68.5 |
| grid | 50.0% | 99 | 53.9 | 54.6 | 62.5 | 62.8 | 68.3 | 68.5 |
| block | 50.0% | 99 | 54.9 | 55.6 | 63.2 | 63.5 | 69.2 | **69.5** |
| **resize** | $160 \times 160$ | 101 | 54.0 | **56.0** | 62.2 | **63.6** | 67.8 | 69.0 |
| random | 75.0% | 50 | 49.5 | 52.7 | 58.51 | 60.9 | 65.9 | 67.6 |
| grid | 75.0% | 50 | 49.5 | 53.1 | 57.9 | 60.7 | 65.4 | 67.3 |
| block | 75.0% | 50 | 45.2 | 53.4 | 57.3 | 61.4 | 65.2 | 68.5 |
| **resize** | $112 \times 112$ | 50 | 50.1 | **54.7** | 59.0 | **62.9** | 65.1 | **68.9** |
| random | 81.6% | 37 | 47.3 | 51.6 | 55.3 | 58.9 | 64.1 | 66.3 |
| grid | 81.6% | 37 | N/A | N/A | N/A | N/A | N/A | N/A |
| block | 81.6% | 37 | 43.6 | 51.7 | 54.8 | 60.7 | 63.1 | 67.6 |
| **resize** | $96 \times 96$ | 37 | 48.3 | **53.9** | 57.0 | **62.1** | 63.8 | **68.1** |
| random | 91.8% | 17 | 36.4 | 47.5 | 44.2 | 55.5 | 55.3 | 62.4 |
| grid | 91.8% | 17 | N/A | N/A | N/A | N/A | N/A | N/A |
| block | 91.8% | 17 | 28.4 | 46.8 | 38.3 | 55.3 | 49.6 | 62.9 |
| **resizing** | $64 \times 64$ | 17 | 40.7 | **50.5** | 51.0 | **59.9** | 58.3 | **66.2** |

Table 11: **Scaling effect on reducing image tokens.** We report top-1 zero-shot accuracy on ImageNet-1k [15] classification. N/A: we adopt 50% and 75% masking ratio for grid mask, larger masking ratio is non-trivial. To ensure a fair comparison, we keep the length of text tokens constant at 32.

| Masking strategy | Image | Text | S/16 | | B/16 | | L/16 | |
|---|---|---|---|---|---|---|---|---|
| | | | pre-train | fine-tune | pre-train | fine-tune | pre-train | fine-tune |
| truncation | $112 \times 112$ | 32 | 50.1 | 54.7 | 59.0 | 62.9 | 65.1 | **68.9** |
| random | $112 \times 112$ | 32 | 50.0 | 54.8 | 59.1 | 62.6 | 65.3 | 68.6 |
| block | $112 \times 112$ | 32 | 50.4 | 54.6 | 59.1 | 62.9 | 65.2 | 68.7 |
| **syntax** | $112 \times 112$ | 32 | 50.1 | **54.9** | 58.7 | 62.6 | 65.0 | 68.3 |
| truncation | $112 \times 112$ | 16 | 50.6 | **55.1** | 58.7 | 62.4 | 65.4 | 68.8 |
| random | $112 \times 112$ | 16 | 49.8 | 54.5 | 58.4 | 62.2 | 65.1 | 68.5 |
| block | $112 \times 112$ | 16 | 50.1 | 54.5 | 59.1 | 63.2 | 65.3 | 68.7 |
| **syntax** | $112 \times 112$ | 16 | 50.2 | 54.7 | 58.9 | 63.0 | 65.3 | **68.8** |
| truncation | $112 \times 112$ | 8 | 45.7 | 54.2 | 54.7 | 62.2 | 62.2 | 68.2 |
| random | $112 \times 112$ | 8 | 44.5 | 53.2 | 54.2 | 61.5 | 61.6 | 67.8 |
| block | $112 \times 112$ | 8 | 45.4 | 54.1 | 54.0 | 62.1 | 61.6 | 68.2 |
| **syntax** | $112 \times 112$ | 8 | 46.7 | **54.6** | 55.6 | **62.9** | 62.3 | **69.0** |
| truncation | $112 \times 112$ | 6 | 30.9 | 52.9 | 39.0 | 60.8 | 47.9 | 67.1 |
| random | $112 \times 112$ | 6 | 299 | 52.1 | 38.4 | 59.7 | 48.0 | 66.8 |
| block | $112 \times 112$ | 6 | 29.3 | 52.9 | 38.3 | 61.0 | 46.8 | 67.8 |
| **syntax** | $112 \times 112$ | 6 | 31.1 | **53.7** | 39.7 | **61.8** | 49.3 | **68.4** |
| truncation | $112 \times 112$ | 4 | 24.1 | 49.0 | 32.6 | 57.7 | 40.4 | 63.6 |
| random | $112 \times 112$ | 4 | 22.0 | 48.9 | 29.3 | 57.1 | 39.6 | 63.6 |
| block | $112 \times 112$ | 4 | 24.0 | 50.3 | 31.5 | 58.8 | 39.6 | 65.8 |
| **syntax** | $112 \times 112$ | 4 | 24.7 | **51.5** | 32.2 | **59.6** | 39.6 | **66.3** |

Table 12: **Scaling effect on reducing text tokens.** We report top-1 zero-shot accuracy on ImageNet-1k [15] classification. **Bold** represents the best performance among different reducing strategies.

**Ablation on fine-tuning schedule and masking.** In addition to random masking, we hereby investigate how grid masking and block masking affect fine-tuning performance. The results are reported in Table 9. Interestingly, compared to fine-tuning input tokens at the full resolution, we observe that 25% masked random fine-tuning and block fine-tuning all lead to a slight performance improvement. With a larger masking ratio, all these masking strategies will lead to worse performance than full-resolution fine-tuning, but overall, random masking consistently yields stronger performance than the other two masking strategies.

|  | ViT-S/16 | | ConvNeXt-T | | ViT-B/16 | | ConvNeXt-B | |
|---|---|---|---|---|---|---|---|---|
| Image size | pre-train | fine-tune | pre-train | fine-tune | pre-train | fine-tune | pre-train | fine-tune |
| 224 × 224 | - | **56.7** | - | 56.7 | - | **64.2** | - | 64.0 |
| 160 × 160 | 54.0 | 56.0 | 55.5 | **56.2** | 62.2 | **63.6** | 63.0 | 63.6 |
| 112 × 112 | 50.1 | 54.7 | 52.6 | **55.3** | 59.0 | 62.9 | 61.0 | **63.1** |
| 96 × 96 | 48.3 | 53.9 | 51.0 | **54.6** | 57.0 | 62.1 | 59.2 | **62.2** |
| 64 × 64 | 40.7 | 50.5 | 44.9 | **51.1** | 51.0 | 59.9 | 54.6 | **60.0** |

Table 13: **Comparison of ConvNeXt [32] and ViT [18].** We report top-1 zero-shot accuracy on ImageNet-1k [15] classification. To ensure a fair comparison, we keep the length of text tokens constant at 32 and only vary the visual backbones. The associated text encoders are specified in Tab. 4.

We next ablate different fine-tuning setups and summarize the results in Table 10. We choose 30% masked random fine-tuning as the default strategy, as it leads to a slight performance improvement (+0.1%) and enables a $1.3\times$ speedup of the fine-tuning process. Furthermore, adopting a $4\times$ fine-tuning schedule results in an additional improvement of 0.6%. However, we empirically find that further increasing the fine-tuning schedule does not lead to any substantial performance gains.

Following [24], we also investigate progressively fine-tuning with large image resolutions. Initially, for the first 512 million samples, we fine-tune the model using a $224 \times 224$ input size with a masking ratio of 30%; subsequently, for the remaining 128 million samples, we adopt a larger $336 \times 336$ input size with a masking ratio of 40% and a smaller learning rate. As shown in the last row of Table 10, *i.e.*, case (5), progressive fine-tuning results in a slight performance improvement of 0.2% compared to direct fine-tuning with a $336 \times 336$ input size and meanwhile achieving a notable $1.5\times$ speedup of the fine-tuning process.

| model | NegCLIP | VG-Relation | VG-Attribute | COCO-Order | Flickr30k-Order |
|---|---|---|---|---|---|
| OpenCLIP-B/16 |  | 44.7 | 59.9 | 41.8 | 45.3 |
| OpenCLIP-B/16 | ✓ | 78.6 (+33.9) | 69.5 (+9.6) | 87.6 (+45.8) | 89.1 (+43.8) |
| CLIPA-B/16 |  | 43.8 | 57.1 | 37.8 | 39.1 |
| CLIPA-B/16 | ✓ | 78.5 (+34.7) | 68.0 (+10.9) | 86.1 (+48.3) | 87.9 (+48.8) |

Table 14: Comparison on ARO benchmark.

## B.3 ImageNet-1k

**Image.** For reference, Tab. 11 presents the numerical zero-shot ImageNet-1k top-1 accuracy of Fig. 4 with various token length reduction strategies. We can see that fine-tuning plays a crucial role with reduced input token length during pre-training, by comparing the performance of pre-trained and fine-tuned models. For instance, fine-tuning pre-trained models with only 17 tokens (the last four rows in Tab. 11) leads to significant performance gains of **18.4%**, **17.0%**, and **18.6%** across S/16, B/16, and L/16 scales, respectively, for block masking.

**Text.** For reference, Tab. 12 presents the numerical zero-shot ImageNet-1k top-1 accuracy of Fig. 7 with various token length reduction strategies. We standardize the visual input size to $112 \times 112$ pixels for all models and vary only the text input during pre-training. Our fine-tuning procedure follows the same approach as that described for the image input. To ensure a fair comparison, we employ the same masking strategy during fine-tuning as used during pre-training when comparing different masking strategies. Note that the pre-training performance is noticeably lower for input texts with a length smaller than 8. This is because the prompt templates we used for evaluation are often longer than a text length of 8. However, after fine-tuning the model with a maximum length of 32, the models performances are significantly improved.

**ConvNeXt [32].** In Tab. 13, we compare the performance of different visual backbones, ViT [18] and ConvNeXt [32]. Notably, we observe that when comparing pre-training results, ConvNeXt outperforms ViT with smaller input sizes, suggesting that CNNs may exhibit greater robustness with respect to scale. For instance, at an input size of $64 \times 64$, ConvNeXt-B outperforms ViT-B by approximately 3.5%. However, after fine-tuning, we observe that the performance gap between the two models narrows considerably across all scales.

## B.4 Zero-shot retrieval and robustness

**Text.** For reference, we also report the performance of zero-shot image/text retrieval on the COCO dataset [30]and zero-shot robustness on the ImageNet-V2 [47], ImageNet-R [22], ImageNet-A [23], and ImageNet-Sketch [58] dataset in Tab. 15 when varying input text token lengths.

| | | | S/16 | | | | | | | B/16 | | | | | | | L/16 | | | | | | |
|---|---|---|---|---|---|---|---|---|---|---|---|---|---|---|---|---|---|---|---|---|---|---|---|
| masking strategy | image | text | IN-1k | IN-V2 | IN-A | IN-R | IN-S | I-to-T | T-to-I | IN-1k | IN-V2 | IN-A | IN-R | IN-S | I-to-T | T-to-I | IN-1k | IN-V2 | IN-A | IN-R | IN-S | I-to-T | T-to-I |
| truncation | 112 × 112 | 32 | 54.7 | 47.1 | 14.4 | 63.0 | 40.1 | 31.3 | 47.7 | 62.9 | 54.7 | 25.2 | 73.6 | 48.4 | 36.6 | 55.3 | **68.9** | 60.5 | 36.3 | 80.8 | 55.5 | 41.5 | 59.9 |
| random | 112 × 112 | 32 | 54.8 | 46.8 | 14.1 | 63.3 | 40.7 | 30.7 | 48.0 | 62.6 | 54.8 | 25.3 | 74.0 | 48.8 | 37.0 | 54.4 | 68.6 | 61.3 | 36.2 | 80.2 | 55.3 | 41.2 | 59.0 |
| block | 112 × 112 | 32 | 54.6 | 47.3 | 13.9 | 62.9 | 40.6 | 31.1 | 48.0 | **62.9** | 54.6 | 25.9 | 74.0 | 49.0 | 36.7 | 54.2 | 68.7 | 60.8 | 36.7 | 81.0 | 55.5 | 41.5 | 59.2 |
| **syntax** | 112 × 112 | 32 | **54.9** | 46.6 | 14.2 | 63.4 | 41.1 | 30.9 | 49.2 | 62.6 | 54.8 | 25.6 | 73.5 | 48.3 | 37.2 | 55.3 | 68.3 | 60.5 | 35.3 | 80.7 | 55.4 | 41.3 | 59.7 |
| truncation | 112 × 112 | 16 | **55.1** | 47.2 | 14.3 | 63.0 | 40.8 | 31.1 | 47.8 | 62.4 | 54.1 | 25.7 | 73.5 | 49.2 | 36.8 | 54.9 | 68.8 | 61.3 | 37.1 | 80.3 | 55.4 | 41.3 | 59.8 |
| random | 112 × 112 | 16 | 54.5 | 46.9 | 14.5 | 63.2 | 40.3 | 30.3 | 48.8 | 62.2 | 54.8 | 25.3 | 73.2 | 48.60 | 36.6 | 54.7 | 68.5 | 60.6 | 37.2 | 81.3 | 55.9 | 40.7 | 59.7 |
| block | 112 × 112 | 16 | 54.5 | 46.9 | 14.5 | 62.6 | 40.2 | 31.1 | 49.1 | **63.2** | 55.3 | 25.6 | 73.3 | 48.6 | 37.1 | 55.0 | 68.7 | 60.0 | 37.2 | 80.5 | 55.7 | 41.2 | 59.4 |
| **syntax** | 112 × 112 | 16 | 54.7 | 47.4 | 13.4 | 62.7 | 40.5 | 30.8 | 48.5 | 63.0 | 55.4 | 25.5 | 73.5 | 49.0 | 37.1 | 55.5 | **68.8** | 61.4 | 37.6 | 80.7 | 55.6 | 41.1 | 58.6 |
| truncation | 112 × 112 | 8 | 54.2 | 46.0 | 14.4 | 63.5 | 40.1 | 29.7 | 47.8 | 62.2 | 54.9 | 25.7 | 74.1 | 48.3 | 35.5 | 53.1 | 68.2 | 61.0 | 37.0 | 80.5 | 55.8 | 39.6 | 56.7 |
| random | 112 × 112 | 8 | 53.2 | 45.6 | 13.7 | 62.6 | 39.1 | 28.6 | 47.1 | 61.5 | 53.5 | 24.8 | 72.6 | 47.2 | 34.6 | 52.6 | 67.8 | 59.7 | 36.2 | 80.1 | 55.1 | 37.8 | 55.7 |
| block | 112 × 112 | 8 | 54.1 | 46.8 | 13.8 | 62.8 | 40.0 | 30.1 | 47.9 | 62.1 | 54.2 | 25.0 | 73.4 | 47.7 | 35.5 | 54.7 | 68.2 | 60.8 | 37.1 | 80.9 | 55.8 | 40.4 | 57.5 |
| **syntax** | 112 × 112 | 8 | **54.6** | 46.6 | 13.6 | 63.5 | 40.1 | 29.9 | 47.7 | **62.9** | 54.7 | 25.7 | 73.4 | 49.1 | 35.4 | 53.5 | **69.0** | 61.6 | 39.1 | 81.1 | 56.3 | 39.7 | 57.7 |
| truncation | 112 × 112 | 6 | 52.9 | 45.9 | 13.3 | 63.6 | 39.3 | 28.5 | 45.9 | 60.8 | 53.5 | 24.2 | 73.2 | 47.7 | 34.4 | 51.4 | 67.1 | 59.5 | 36.3 | 80.5 | 54.8 | 38.2 | 54.9 |
| random | 112 × 112 | 6 | 52.1 | 44.4 | 12.1 | 61.5 | 38.0 | 27.4 | 44.8 | 59.7 | 51.9 | 22.1 | 71.8 | 47.0 | 32.6 | 50.6 | 66.8 | 59.4 | 35.2 | 79.7 | 54.5 | 36.9 | 54.3 |
| block | 112 × 112 | 6 | 52.9 | 45.6 | 13.6 | 62.2 | 39.0 | 29.0 | 45.8 | 61.0 | 53.4 | 24.1 | 72.7 | 47.5 | 34.2 | 51.6 | 67.8 | 60.8 | 36.3 | 80.1 | 55.1 | 39.2 | 57.2 |
| **syntax** | 112 × 112 | 6 | **53.7** | 45.8 | 13.3 | 63.7 | 39.8 | 28.8 | 46.4 | **61.8** | 54.2 | 25.3 | 74.0 | 48.4 | 34.3 | 52.4 | **68.4** | 61.0 | 37.6 | 81.5 | 56.3 | 38.5 | 56.4 |
| truncation | 112 × 112 | 4 | 49.0 | 41.6 | 12.2 | 61.5 | 36.3 | 24.9 | 40.6 | 57.7 | 50.2 | 21.3 | 71.8 | 44.8 | 30.6 | 47.2 | 63.6 | 56.3 | 33.6 | 79.1 | 52.9 | 34.2 | 49.8 |
| random | 112 × 112 | 4 | 48.9 | 41.7 | 11.4 | 59.2 | 35.2 | 25.0 | 40.6 | 57.1 | 49.4 | 19.0 | 69.8 | 44.2 | 29.6 | 46.1 | 63.6 | 55.2 | 30.5 | 77.6 | 52.4 | 33.2 | 49.0 |
| block | 112 × 112 | 4 | 50.3 | 42.3 | 12.5 | 60.4 | 37.1 | 26.0 | 41.6 | 58.8 | 50.8 | 21.0 | 71.5 | 46.0 | 31.9 | 49.2 | 65.8 | 57.8 | 33.0 | 79.7 | 54.3 | 35.8 | 52.9 |
| **syntax** | 112 × 112 | 4 | **51.5** | 44.4 | 13.4 | 63.7 | 38.0 | 26.2 | 42.6 | **59.6** | 52.5 | 22.5 | 72.7 | 46.8 | 31.5 | 48.4 | **66.3** | 58.2 | 35.6 | 80.7 | 54.8 | 35.1 | 51.7 |

Table 15: **Zero-shot image/text retrieval and robustness.**We also study the scaling effect on text tokens on image/text retrieval tasks and robustness benchmarks.

