# OpenReview forum: "An Inverse Scaling Law for CLIP Training"
_NeurIPS.cc/2023/Conference — NeurIPS 2023 poster_

### Official Review · Reviewer_Depv · 2023-06-25

**Soundness:** 4 excellent
**Presentation:** 3 good
**Contribution:** 3 good
**Rating:** 7
**Confidence:** 4

**Summary:**

This work explores how to increase the training speed of CLIP in order to make it possible to train on academic resources. They examine how to best compress both the visual and text information given a limited token budget. They do this across multiple CLIP sizes and find that as the size of CLIP increases, it sees less of a drop for fewer tokens which they deem the "Inverse Scaling Law".

**Strengths:**

This is a very useful line of work, as I believe there is still a lot of research to be done in the Language-Vision space and the compute requirements of CLIP is a huge barrier. Allowing CLIP training on an academic resource budget is valuable to the greater resource community.

It is presented in an easily understood way. I especially Like figure 7, which summarizes their results nicely.

**Weaknesses:**

My main concern is novelty. Their findings are very similar to that in "RECLIP: Resource-efficient CLIP by Training with Small Images" which also found that images could be sized down and text could be truncated without major accuracy loss, with the goal of increasing training accessibility. This paper essentially takes the main findings of this paper, and provides extensive ablations for a series of design choices (such has the best way to compress the images/text), but ultimately comes to a very similar conclusion. However, as RECLIP, was only on ArXiv 2 months ago, I believe it should be considered as concurrent work. I would however ask that the authors include a discussion of this work in their next version.

**Questions:**

- How do you see your work in relation to the concurrent RECLIP (https://arxiv.org/abs/2304.06028)?
- What do you see as the main limitations of this work?
- [minor note] typo in line 208 (double ampersands)

**Limitations:**

In future versions I would recommend a section for limitations.

---

> ### Author Rebuttal · Authors · 2023-08-09
>
> We first thank the reviewer for the detailed comments and the appreciation of our work. We address the concerns below:
>
>  ### Q1: Relation to RECLIP
> Thanks for bringing up this concurrent work! We totally agree that a discussion of RECLIP [1] will make this work more complete, and we will do so in the next version. Our work does share some similarities with RECLIP. For example, both aim to ease the computational burden of CLIP training. And both propose image resizing as an effective token length reduction strategy. However, we would like to highlight our unique contributions as follows.
>
> Beyond RECLIP, our work takes one step further, aiming to investigate how the performance of different model scales responds to reduced-length input.  By exploring eight image/text token reduction strategies, we uncover the inverse scaling law, a novel observation that serves as a vital guide for accelerating CLIP training in general. Furthermore, we extensively validate the effectiveness of our approach with a wide range of compute budgets (ranging from 18 to 370 GPU days), model scales (from S/16 to H/14), training datasets (LAION400M, LAION2B, and DataComp-1B), evaluation benchmarks (Robustness, Retrieval, and VTAB), demonstrating the well-established generalizability of our work.
>
>
>
>  ### Q2: Main limitation
> Our work's primary limitation is that we only focus on the inverse scaling law in CLIP training. While this has led to promising insights, we have yet to explore its applicability to other image-text learning tasks. Given that 1) both image and text data often contain (a lot) redundant information; and 2) larger models have stronger representation learning ability, we believe future endeavors in this direction to be potentially fruitful. This exciting direction for further exploration is left as future work.
>
>
>  ### Q3: Typo
> Thanks for pointing out the typo! We will correct it and further polish the writing of this paper.
>
>
> **References**
>
>  [1] Li, Runze, et al. "RECLIP: Resource-efficient CLIP by Training with Small Images." arXiv preprint arXiv:2304.06028 (2023).

---

> > ### Comment · Reviewer_Depv · 2023-08-12
> > **Thanks for your response.**
> >
> > I've increased my score.

---

> > > ### Author Response · Authors · 2023-08-20
> > >
> > > We are glad to see all your concerns have been addressed! Thanks for increasing the score!

---

### Official Review · Reviewer_TeGF · 2023-07-02

**Soundness:** 4 excellent
**Presentation:** 3 good
**Contribution:** 3 good
**Rating:** 8
**Confidence:** 5

**Summary:**

Different methods to reduce the number of tokens entering the image or text encoders when training a CLIP model are compared.
In doing this the authors discover that larger models are able to learn faster from fewer tokens per example than smaller models; this is the inverse scaling law in the title.

Models are compared keeping the following variables constant:

* Training set: LAION-400M
* Total pre-training examples: 6.4 epochs
    * Tokens/example is **not** constant so tokens/epoch will be vary accordingly
* Total fine-tuning examples: 0.36 epochs
* Learning rate: 8e-6 pre-training and 4e-7 fine-tuning
    * However, this is not kept constant for S/16 and B/16 models in scaling experiments

The authors investigate alternative strategies to reduce tokens/example based on FLIP, designing them based on the attributes of images (grid masking: "preserving one patch in each 2x2 grid window") or attributes of text (syntax masking preferentially retains nouns). They then investigate the effect of scaling while reducing tokens/example.

Using the discovery that larger models learn faster from fewer tokens the authors are able to train a CLIP Academic (CLIPA) (I can't find where this is defined in the paper but I think this is what it is?) on LAION-2B using 1/15 the compute budget of OpenCLIP at the H/14 scale and achieve comparable performance.

**Strengths:**

The experiments in this paper are comprehensive and follow a reasonable thread of investigation.
The authors define clearly what they aim to investigate, then show complete results of that investigation, then apply those results to a pressing problem in the field.

The comparison of different methods to reduce tokens/example is valuable and the methods chosen for comparison seem like good choices. The results are presented in full in Figures 5 and 6 and Table 1.

The training results are striking and of obvious importance to the community. CLIP was a major step in the field of deep learning so speeding up training is valuable to increase access and to speed up experimental iteration on these architectures. The speedup of 16x is more than an order of magnitude.

This paper builds on the results of FLIP by making a useful observation on the effect of model scaling on token/example reduction strategies.

**Weaknesses:**

Figure 1 is difficult to parse because the number of tokens in the top row could indicate entries in the contrastive matrix or it could indicate the number of tokens at the input to the image/text encoders. From reading the paper, I think it is the latter but it would be helpful if this was immediately obvious. If this mirrored the design of Figure 1 in the original CLIP paper I think it would be very clear.

The results of this paper are quite different if we view epochs as being a measure of tokens or of steps. I think that it is a measure of steps because the alternative would be more difficult to code but I think it would be worth making clear so that's what I wrote in the summary above. I think to make this extremely clear and better state the scaling law there should be a figure comparing log total training tokens to loss or accuracy. From Figure 4 I think this would show increasing accuracy with the number of tokens processed by the network during pre-training.

It is not clear in Section 5 exactly which methods are applied in CLIPA training to reduce tokens/example, from the methods proposed in previous sections. It would be useful for this to be made clear, even if it is stated in prior sections or appendices, because the results hinge on this.

The speedup quoted in Section 6 and in line 51 compares to OpenCLIP but Table 4 shows similar comparison to the work this is most directly based on, FLIP, was trained with only 5x the compute. This is a more fair comparison and deserves to be stated instead of the alternative unless you qualify that it is 16x cheaper than training without doing anything to reduce tokens/example.

Smaller issues:

* In the abstract the bold results at the end are impressive only if one knows the training time of OpenCLIP but it might be more reasonable to assume that most readers won't
* Lines 33-36 describe the idea of a scaling law in parameters but neglect scaling laws in tokens or compute etc, it might be valuable to be more precise
* The text in Figure 5 x-axes is very small, could it be made more legible somehow?
* I think Figure 7 shows the effect of the tokens/example reduction strategies on GFLOPS required for training but the caption does not explain what is happening in this Figure. Is GLOPS the number of flops per example, so the x-axis would show the total training cost?

**Questions:**

What is the formal definition of a scaling law this paper is applying, such that this demonstrates an inverse scaling law? I'm not sure if the field has one, it seems to be "a straight line predicting performance on a y-axis and training cost on a logarithmic x-axis".

Have I misunderstood the role of the masking methods to reduce tokens/example? I don't think I have but my entire review depends on it.

**Limitations:**

Comparing this to a [Chinchilla][] scaling law framework it may be that CLIP models are typically trained with more tokens than necessary, in which case we would expect to observe larger models perform better than smaller models regardless of the reduction in tokens. This could be investigated in a way analogous to the Chinchilla paper, by varying the total number of training tokens seen be the model for different model scales and aiming to see the IsoFLOP curves shown in Figure 3 of the Chinchilla paper. However, this would be a computationally expensive experiment to run. Perhaps the authors can come up with a cheaper way?

[chinchilla]: https://arxiv.org/abs/2203.15556

---

> ### Author Rebuttal · Authors · 2023-08-09
>
> We first thank the reviewer for the detailed comments and the appreciation of our work. We address the concerns below:
>
> ### Q1: Figure 1 difficult to parse
> Thanks for raising the concern! The number of tokens in the top row indeed indicates the number of input tokens. We will update Figure 1 to make it clearer in the next version.
>
> ### Q2: Epochs as a measure of steps or tokens
> Thanks for this suggestion! As requested, we've included an alternative view in the rebuttal (Figure 3) that compares total pre-training tokens to accuracy, and will include it in the next version.
>
> ### Q3: Token reduction strategy used in CLIPA
> Thanks for the suggestion! Image resizing is used in all CLIPA training. As for text masking, it depends on model size. Specifically, for small models (S/16, B/16 and L/16), as the text encoder incurs only a small portion of the total compute, we follow FLIP’s practice and opt for truncation. However, when scaling to H/14 with 70x70 image input, we note reducing the text token from 32 to 8 results in a substantial reduction of ~40% in compute requirements. We select syntax masking for CLIPA H/14 due to its strong performance. We will make this setup clear in the next version.
>
> ### Q4: Speed comparison with FLIP in section 6
> Indeed, FLIP H/14 was only trained with ~6x compute compared to our CLIPA-H/14 model. This ~15x compute result is based on the OpenCLIP training with full-length input. We will make this speed comparison clearer in the revision.
>
> ### Q5: Bold results in the abstract
> Thanks for the suggestion! We will include background information about the original CLIP training cost in the abstract to better aid readers in comprehending the significance of the results presented.
>
>  ### Q6: Line 33-36 a scaling law in parameters
> Thanks! We will rephrase our statement about the scaling law to make it more precise and accurate.
>
>  ### Q7: Text in Figure 5 axes too small
> Thanks! We will increase the font size in Figure 5 axes for better presentation in the next version.
>
>  ### Q8: Figure 7 caption
> Yes, GFLOPS is the number of flops per example, and the x-axis of Figure 7 shows the total training cost. We will update the caption to make it clearer in the revision.
>
>  ### Q9: Definition of the inverse scaling law
> The common scaling law associates more compute and larger model scale with better performance [1]. The inverse scaling law reveals that larger image/text encoders allow for the use of shorter image/text token sequences in CLIP training. The 'inverse' naming refers to this negative correlation.
>
>  ### Q10: The role of the masking methods
> You understand it correctly! Appropriate input token reduction greatly accelerates CLIP training at little-to-no performance drop. This is the main point of this work.
>
>  ### Q11: Token Redundancy
> Intuitively, we agree with the idea that the redundant information in the image-text pairs is what makes the token reduction strategy work. This is evidenced by our results in Figure 7. The larger models have stronger representation learning capacity, and thus can learn better with input of reduced-token version, surpassing the smaller models trained with full-size input by a significant margin. Further evidence lies in our CLIPA-H/14 model (I36, T8), which achieves performance comparable to OpenCLIP-H/14 using only about 1/15 of the total number of tokens.  We will add a related discussion in the next version.
>
> **References**
>
> [1] Kaplan, Jared, et al. "Scaling laws for neural language models." arXiv preprint arXiv:2001.08361 (2020).

---

> > ### Comment · Reviewer_TeGF · 2023-08-10
> > **Thanks for clarifying**
> >
> > From your rebuttal I think I understood the paper in my review and I think it is a worthwhile submission for the conference. I will update the confidence of my review.

---

> > > ### Author Response · Authors · 2023-08-20
> > >
> > > Thanks for your support! We deeply appreciate your acknowledgment of the significance and potential of this work.

---

### Official Review · Reviewer_KzVG · 2023-07-03

**Soundness:** 2 fair
**Presentation:** 2 fair
**Contribution:** 3 good
**Rating:** 3
**Confidence:** 4

**Summary:**

This paper is about increasing the training efficiency of vision-language models such as CLIP. These models can serve as a foundation for many vision applications and are therefore very useful. However, they are typically (pre-)trained on large datasets for many steps, which makes research involving training such models costly. This motivates efforts to reduce the training cost.

The paper focuses on the sequence length of both the image encoder and the text encoder in CLIP-style VLMs, which is a key determinant of model compute and memory requirements. The paper evaluates random token dropping, dropping tokens in grids or blocks, and reducing the image resolution. The paper finds that reducing image resolution performs best based on zero-shot ImageNet classification performance.

For the text encoder, the paper finds that dropping text tokens such that nouns are preferentially kept and other words are dropped yields the best performance, again on zero-shot ImageNet classification.

The paper also sweeps across different model sizes and finds that larger models can tolerate larger reductions in sequence length on both the image and text encoder.

With the proposed sequence length reductions, it is possible to train models matching or exceeding the performance of OpenCLIP (but not original OpenAI CLIP) on image classification and image/text retrieval in 10-20x less training time.


**Strengths:**

1. The work is well motivated. Reducing the cost of pretraining foundation models has broad impact.
2. The reduction in training time is significant and could make research on training VLMs easier (although it comes with caveats, see below).
3. The presented experiments are clear and and well-structured.

**Weaknesses:**

1. **Incomplete evaluation:** The paper relies on zero-shot classification and retrieval to evaluate the proposed model against the original CLIP and OpenCLIP. This covers only a small part of the tasks that CLIP is typically used for. The original CLIP paper also evaluates representation quality with linear probing and finetuning, on a much wider range of tasks and image types. These evaluations may be more informative about general representation quality than zero-shot image classification. In particular, approaches such as reducing the input resolution and dropping non-noun words may work for classification, where there is a single big object in the image and the text usually consists of a single noun, but might harm more structured or fine-grained downstream applications. This should at least be discussed and ideally tested, e.g. with linear probing and finetuning, and a wider range of tasks, ideally including structured tasks such as segmentation or detection.

2. **Inaccurate and unnecessarily polemic presentation:** The paper makes several inaccurate statements with the apparent motivation of drawing attention. These inaccuracies distract from the content:
    1. **Inverse scaling law:** Despite the title, the paper does not propose a quantitative scaling law, and there is nothing “inverse” about the presented scaling results. Figure 7 shows that as the compute budget increases, the optimal model size grows. This is the same result first presented in Figure 2 of https://arxiv.org/pdf/2001.08361.pdf and confirmed in many works since. The observation that larger models tolerate larger reductions in sequence length neither makes the scaling “inverse” nor is it a surprising result, since bigger models are known to be more sample efficient (https://arxiv.org/pdf/2001.08361.pdf, https://arxiv.org/pdf/2106.04560.pdf), i.e. they learn faster from less data than smaller models. I suggest rephrasing the title to avoid confusion.

    2. **CLIP restricts accessibility:** The abstract and introduction suggest that foundation models such as CLIP are “restricting accessibility to a small group of researchers and technology companies”. This seriously misrepresents the impact these models had on low-resource research. In fact, foundation models like CLIP made it possible to train a model once and then apply it zero-shot, or with cheap fine-tuning, to many different tasks. Sharing pre-trained foundation models therefore has been a huge gift to low-resource research, since complex research questions can now be explored with a pretrained CLIP model, often without having to do any model training. Portraying such models as hindering low-resource research is inaccurate and unfair towards the researchers who develop and release them. I suggest the authors revise these statements.

    3. **Academic resources:** Another questionable premise is that of “academic” resources (e.g. in the title of Section 5 and the name of the proposed method). There are academic labs with large resources and industry labs with low resources. In addition, research labs in industry that publish their work can reasonably be considered to be part of academia (for reference, Wiktionary defines “academia” as “The scientific and cultural community engaged in higher education and research, taken as a whole.”). Equating “academic” with  “low resource” is therefore inaccurate and reinforces stereotypes. Please simply talk about “few/low resources” instead of “academic resources.”

In summary, additional evaluation and major revision of the presentation are necessary before publication at NeurIPS.

**Questions:**

Further suggestions:
* Please provide Table 1 data in graphical form like Figure 4.
* Table 3 should include inference FLOPs to be able to compare models.

Typos:
* L170: For instance, For instance
* L208: Double ampersand

**Limitations:**

The paper does not address the concern that the limited evaluation of the proposed model may hide model deficiencies compared to the original CLIP. This should be addressed in the rebuttal.

---

> ### Author Rebuttal · Authors · 2023-08-09
>
> We first thank the reviewer for the detailed comments. We address the concerns below:
>
> ### Q1: Incomplete evaluation
> Thanks for raising this concern!
>
> Firstly, it is essential to note that our training pipeline contains two stages — the reduced-length input is only applied during pre-training to enable fast CLIP training, and full-length input is used during fine-tuning to ensure that models learn from the complete information available in the training data.
>
> Secondly, as suggested by Reviewer UF9M, for a wider range of evaluation, we have included the results on VTAB benchmarks in the rebuttal file. The competitive performance of our CLIPA models on this diverse and comprehensive benchmark demonstrates the good generalizability of CLIPA.
>
> Lastly, we argue that, compared to other learning paradigms, the biggest advantage of CLIP lies in its extraordinary zero-shot recognition ability. This is also highlighted in both the official CLIP repo and the popular OpenCLIP repo. Our work follows this common practice and our assessment of models’ zero-shot ability on a wide range of datasets demonstrates the efficiency and the effectiveness of our CLIPA.
>
> We will add these clarifications in the next version.
>
>
> ### Q2: Inverse scaling law
> We first stress that our observation indeed suggests an inverse relationship between the model size and the length of image/text tokens in CLIP training — that the larger the image/text encoders used, the shorter the sequence length of image/text tokens that can be applied in training.
>
> Secondly, it's crucial to distinguish our findings from the scaling law in [1], though seemingly subtle. The sample efficiency in [1] highlights that in language learning, larger models need fewer samples to achieve the same performance as the smaller models. This concept could be applied to CLIP training (e.g., L/16 may need ~1/3  total input samples to reach the same performance as S/16), but this is NOT what we discussed/explored in this work. Instead, we argue that, compared to small models, we could drop more information **per training sample** to train large models (e.g., for a performance drop threshold of ~1%, S/16 needs 160x160 image input, but L/16 only needs 96x96 image input).
>
> Therefore, while both ours and [1] demonstrated that large models could be trained more efficiently, the specific scenarios and the underlying rationale are distinct, i.e., [1]’s token efficiency is realized by using few training samples (i.e., there is no token drop in each training sample), but our token efficiency is realized by using fewer tokens in each training sample (i.e., total training samples keep the same). To our knowledge, this specific observation is novel and unexplored in prior works.
>
> We will make these points clear in the next version.
>
>
>
>
> ### Q3: CLIP accessibility statement inappropriate
> Sorry for the confusion. First, we agree that, by building upon open-sourced CLIP models, simple zero-shot applications or cheap fine-tuning of CLIP have made significant contributions to the field and led to many fruitful research outcomes.
>
> However, our comments on accessibility were specifically targeted at the training aspect of CLIP. While the release of the OpenCLIP codebase and the LAION datasets (400M and 5B) has indeed empowered researchers to train their own CLIP models, the substantial training cost associated with the large-scale model and data remains a significant barrier to further exploration.
>
> Our intention was to highlight this aspect of the accessibility challenge, as we believe that there is still ample room for community-driven innovation in improving or even completely revamping the image-text learning paradigm. CLIP's current form, though powerful, is likely not the final solution.
>
> Our discovery of the inverse scaling law addresses this challenge by enabling CLIP training with significantly reduced costs and faster research cycles. By making training more feasible, we hope to stimulate greater participation and creativity in the field.
>
> We will make this point clearer in the next version.
>
>
>
> ### Q4: Academic resources statement inappropriate
> Thanks for the suggestion. We will revise our language to more precisely reflect our focus on low-resource scenarios (e.g., a standard 8-GPU server).
>
>
> **References**
>
> [1] Kaplan, Jared, et al. "Scaling laws for neural language models." arXiv preprint arXiv:2001.08361 (2020).

---

> > ### Comment · Reviewer_KzVG · 2023-08-15
> >
> > Thanks for your response.
> >
> > Q1: Thanks for providing the VTAB results. They do not address my concern that the short token sequences may bias CLIPA so that it does well on classification, but perhaps less well on tasks requiring more complex language understanding. VTAB is a classification benchmark, so it adds little additional information here. In fact, as Reviewer UF9M pointed out, CLIPA performs worse on retrieval, which may be due to the shortened sequences during (pre-)training.
> >
> > Q2. The authors write "it's crucial to distinguish our findings from the scaling law in [1]". One way to distinguish their work from [1] would be to not appropriate the term "scaling law". What the paper presents is simply not a scaling law in the sense commonly used in the literature (as other reviewers have also pointed out). Regarding the term "inverse": A simple way to re-state your findings would be to say that as models get bigger, performance goes up. When performance goes up, you can train on less data (or shorter sequences) to reach the same performance. So there's nothing inverse -- it's just a re-statement of the well-known fact that bigger models reach higher performance. It is therefore an exaggeration to talk about a new "discovery" here, since the method is an application of well-known facts about neural networks.
> >
> > My original review continues to apply.

---

> > > ### Author Response · Authors · 2023-08-20
> > >
> > > Thank you for the additional comments.
> > >
> > > ### Q1: Language Understanding
> > > Thanks for raising this concern. In light of your suggestion and as recommended by Reviewer Mhr1, we delved into the ARO benchmark to further our understanding. In short, our results 1) echo the finding in [1] that **CLIP models generally exhibit limitations in capturing relationships, attributes, and order information** and 2) confirm that NegCLIP, developed in [1], can effectively serve as a lightweight and drop-in solution to enhance the text understanding ability of both OpenCLIP and CLIPA. For a comprehensive breakdown of our observations on this front, please refer to our response to Reviewer Mhr1’s Q1.
> > >
> > > In addition, we would like to clarify the concerns about the drop in text retrieval performance (presented in Table 3). In short, our CLIPA models in Table 3 are intentionally trained with limited resources (e.g.  on a 8-GPU server). Considering that other baselines use **20-60x** more compute, the fact that our CLIPA achieves competitive results is noteworthy, and should be considered as one of our key contributions. In addition, a strong evidence is that, when we scale to the size of H/14, our CLIPA model exhibits competitive outcomes in both classification and retrieval tasks compared to OpenCLIP (as presented in Table 4). For a more nuanced understanding of this aspect, please refer to our response to Reviewer UF9M’s Q2.
> > >
> > >
> > >
> > > ### Q2: Inverse Scaling Law
> > > We’d like to first reiterate that our findings are **novel** and should be distinguished from the scaling law in [2]. The differences can be summarized as:
> > >
> > > 1) the sample efficiency in [2] is realized by using few training samples, but our token efficiency is realized by using fewer tokens in each training sample.
> > > 2) our inserve scaling law focuses on **performance drop**, which really is a comparison between the **same** model trained with reduced-length or full-length input. One illustrative example is our CLIPA-H/14 (I36,T8), which is trained with **~87%** less tokens per sample during pre-training, and yet shows *barely any performance drop* compared to OpenCLIP-H/14 trained with full-length input. This phenomenon is not revealed in prior scaling law works.
> > >
> > > Additionally, we stress that our results indeed present an inverse relationship between the model size and the length of image/text tokens in CLIP training — that the larger the image/text encoders used, the shorter the sequence length of image/text tokens that can be applied in training.
> > >
> > > We will make these points clear in the next version.
> > >
> > >
> > > **References**
> > >
> > > [1] Yuksekgonul M, Bianchi F, Kalluri P, et al. When and Why Vision-Language Models Behave like Bags-Of-Words, and What to Do About It?[C]//The Eleventh International Conference on Learning Representations. 2022.
> > >
> > > [2] Kaplan, Jared, et al. "Scaling laws for neural language models." arXiv preprint arXiv:2001.08361 (2020).

---

### Official Review · Reviewer_UF9M · 2023-07-05

**Soundness:** 3 good
**Presentation:** 2 fair
**Contribution:** 3 good
**Rating:** 4
**Confidence:** 4

**Summary:**

This work studies the impact of token length (image and text) on the quality of image-language pretraining in CLIP-style models. Crucially, the authors find that as the model size (both image and language encoders) are scaled up, the drop in accuracy owing to a smaller input token length keeps on decreasing. This is pitted as the inverse scaling law. A smaller input token size enables training significantly faster, achieving nearly 15x speed up over the conventional training protocols, and also improving over the gains observed in prior work, FLIP.

**Strengths:**

1. The work provides a comprehensive analysis of various token reduction strategies for both image and text and also proposes new ones.
2. Analysis of different scales of models, across different masking strategies is really helpful to convert a previously under-explored finding into a consistent phenomenon that can be useful for future research.
3. The work makes large model training accessible by suggesting an alternate training paradigm of pre-train and then fine-tune that enables CLIPa training at a much lower cost than before (15x speedup).

**Weaknesses:**

1. I find Figure 1 confusing. What is the relation between the encoder input token length and the encoder output "image/text embedding" dimension. This should be discussed in more detail in the paper. Also, since the whole work is about token length reduction, I think it is important for the authors to discuss how images are converted into patches and then into tokens and how this calculation is done.
2. I do not feel very confident about the results in the paper because the parameter choices appear to be heavily tuned in order to get good performance for the chosen strategy of resizing. For instance, the authors use a base learning rate of 8e-6 in the main training stage and 4e-7 for fine-tuning---these are non standard parameters and a discussion on parameter tuning is important here.
3. Similarly, in line 147 the authors write that "one exception that a larger learning rate of 8e-7 is utilized during fine-tuning for S/16 and B/16." Such opportunities were not given to baselines such as FLIP. Further, special tricks and augmentations were used for H-14 making it unclear how the results will translate further.
4. Scaling Law: Since this paper advertises a scaling law, the graphs should have model size on x axis and the various fractions of token reduction as individual lines. Moreover, there should be more points on the "new" x axis to claim a scaling law.
5. Why are there no graph for results corresponding to Table 1.
6. Evaluations: Model performs significantly worse on text retrieval tasks. I think the authors should do a more comprehensive evaluation of CLIPA on various benchmarks such as VTAB (check datacomp paper for current best practices).
7. Table 3 should compare with FLIP and other masking strategies






**Questions:**

1. What is the performance of CLIPA models without fine tuning
2. How was the fine tuning duration and learning rate chosen?
3. How were parameters for pretraining chosen?
4. What are the embedding dimensions of the models when the token lengths are changes (see Figure 1)

**Limitations:**

Discussed in Weaknesses

---

> ### Author Rebuttal · Authors · 2023-08-09
>
> We first thank the reviewer for the detailed comments. We address the concerns below:
>
> ### Q1: Figure 1 confusing & Discussion about patchifying
> Sorry for the confusion about Figure 1. There is a typo in the figure; the dimension of the encoder output solely depends on the encoder size (i.e., a larger encoder size leads to a higher output dimension) and is unrelated to the input token length. We will fix it.
>
> Regarding patchifying process, we follow standard ViT practice to partition an image into $p×p$ (i.e., p=16 or 14 in our experiments) non-overlapping patches and project each patch into a token. To reduce the token length, we apply techniques such as random, block, or grid masking to remove tokens. When resizing, the input size diminishes, but the patchify size remains consistent, resulting in a reduced token length.  We will clarify it in the next version.
>
>
> ### Q2 & Q3: Heavy & Unfair parameter tuning
> We appreciate your concern about potential heavy or unfair parameter tuning. Nevertheless, we wish to clarify that our approach is not characterized by such practices. Our training setup predominantly aligns with FLIP's, with the only exception being the use of a slightly higher learning rate. It's also important to emphasize that our random masking baseline corresponds to FLIP and is trained using this new learning rate configuration. As our analysis in Section 3 indicates, random masking, as implemented in FLIP, is not the optimal solution for reducing token length in CLIP training.
>
> Furthermore, we emphasize that the key contribution of this paper is the discovery of the inverse scaling law in CLIP training. We firmly believe that this finding is obtained under a fair experiment setup.
>
> ### Q4: Scaling law figure drawing
> Thanks for the suggestion. Please see this new figure ‘with model size on the x-axis and the various fractions of token reduction as individual lines’ in the rebuttal file (Figure 2). We will add it to the next version to clarify the observed scaling law better.
>
> ### Q5: Graph for Table 1
> We have included this alternative view for Table 1 in the rebuttal file (Figure 1) and will include them in the revision.
>
> ### Q6: A more comprehensive evaluation
> Thanks for your suggestion. As requested, we compare OpenCLIP, FLIP, and our CLIPA models on VTAB benchmarks. The results are included in the rebuttal file, as shown in Table 1. As can be seen, on this highly diverse and challenging set of vision tasks, our CLIPA still achieves comparable or even superior performances but with significantly less training cost, demonstrating our approach's good generalizability. We will include these results in the next version.
>
> ### Q7: Include other masking strategies in Table 3
> The detailed comparisons among different masking strategies (including FLIP/random masking) are shown in section 4 (Figures 4-6). Given these findings, we made the decision to focus on the most effective strategies (i.e., resizing for image, syntax masking for text) when scaling to large models like H/14. We will make it clear in the next version.
>
> ### Q8: CLIPA performance without fine-tuning
> We reported the performances of pre-trained CLIPA models in the supplemental material shown in Tables 4, 5 & 6. We will make it clear in the next version.
>
> ### Q9 & Q10: Choosing parameters for fine-tuning & pre-training
> Our pre-training and fine-tuning duration exactly follows the FLIP setup (in pre-training, 6.4 epochs for ablation and 32 epochs for scaling; 0.32 epochs in fine-tuning). The learning rates, as answered in Q2&Q3, are slightly larger than the FLIP choices. We will make it clearer in the next version.

---

> > ### Comment · Reviewer_UF9M · 2023-08-17
> >
> > I thank the authors for their efforts in responding to the questions.
> >
> > 1. First, the results on VTAB also show that the performance of CLIPA models when compared to OpenCLIP models is quite random, performing worse by quite a few points in many datasets, and also similarly performing better in a few others. Added to this is the significant drop in retrieval performance, which is one of the most important use case of image-language models.
> > 2. I am not satisfied with the answer on hyperparameter tuning. In particular, the authors still do not answer how they selected these parameters.
> > 3. I do not like the framing of this paper trying to pitch the story as an inverse scaling law when results are only discussed at 3 points on the x-axis, and when going to H-14 models special tricks and augmentations need to be used, making it unclear how the results will translate further. I resonate with the opinions of Reviewer KzVG in this regard.
> >
> > I congratulate the authors on their new efforts that have led to a new state of art open-source model, and recognize the efforts that this might have taken. I do think that the claim that "vision embeddings" of CLIP models do not need "long sequence length in text embedding" is the correct framing of this paper and this would still be valuable for publication at NeurIPS. But in its current form, the claims are an overstatement.

---

> > > ### Author Response · Authors · 2023-08-20
> > >
> > > Thank you for these additional valuable suggestions, which are important for furthering the quality of our work.
> > >
> > >
> > > ### Q1: VTAB result
> > > When examining the VTAB benchmarks, our CLIPA-H/14 model outperforms on 6 datasets, the OpenCLIP-H/14 excels in 4, and FLIP leads in 6 datasets. This competitive performance of CLIPA, especially when considered alongside its efficiency benefits, substantiates our rebuttal claim that “our CLIPA still achieves comparable or even superior performances”
> > >
> > >
> > > ### Q2: Drop in text retrieval performance
> > > We recognize the concern raised about the drop in text retrieval performance (presented in Table 3). However, it is essential to contextualize our work's primary objective, which is uncovering the inverse scaling law in CLIP training — i.e., The performance drop caused by shorter token length will diminish for larger models. This observation is substantiated by our results presented in Table 4. For instance, when we scale to the size of H/14, our CLIPA model exhibits competitive outcomes in both classification and retrieval tasks
> > >
> > >
> > > Moreover, the intent behind Table 3 is not merely to showcase the absolute performance of CLIPA, but to demonstrate its potential in constrained resource settings. We intentionally trained CLIPA with strict computational bounds, limiting the training duration to 2-4 days on an 8-GPU A100 server. When considering these constraints, the fact that CLIPA can achieve competitive results is noteworthy. While other baselines outperform due to their **20-60x** higher computational budget, the efficiency and competitive performance of CLIPA in its restricted setting are pivotal aspects of our contribution.
> > >
> > >
> > > ### Q3: Hyper-parameter tuning
> > > Compared to FLIP, the only hyperparameter change in training is learning rate. The main reason is that our total training epoch is only ~6.4 epochs (which is relatively short, making models under-trained), therefore a slightly larger learning rate is applied to help convergence.
> > >
> > >
> > > It is crucial to note that our primary conclusion remains unaffected by this hyperparameter change. For clarity, 1) when applying the short 6.4 training epochs, all our baselines (including random masking/FLIP) are trained under the same hyper-parameter setup, therefore ensuring fair comparisons; and 2) for public baselines (like OpenCLIP) which are trained with sufficient epochs (i.e., 32 epochs), we verified that this learning rate change only lead to a negligible performance variation (e.g., within 0.2% in zero-shot ImageNet classification).
> > >
> > >
> > > ### Q4: Scaling law
> > > We stress that our observed scaling law in CLIP training is different from the typical scaling law observed in language models. More specifically, what we are trying to clarify is that larger models can apply shorter token length in CLIP training. We extensively examine this scaling law with 8 different strategies, each at varying masking ratios. By ranging the model size from S/16 to L/16, we obtain a total of **105** data points in well supporting the conclusion of inverse scaling law in CLIP training.

---

### Official Review · Reviewer_Mhr1 · 2023-07-07

**Soundness:** 3 good
**Presentation:** 3 good
**Contribution:** 3 good
**Rating:** 6
**Confidence:** 4

**Summary:**

The key finding in this paper is that, the larger the model is, the shorter the token sequence is required to maintain a certain level of performance. Based on this finding, this paper proposes to train larger model with shorter sequences, which is more computationally favorable than training smaller models with sequences of full length. As a result, one can achieve 69.3% top-1 zero shot image accuracy with just eight A100 GPUs in ~4 days.



**Strengths:**

- The finding about the inverse scaling law for CLIP is interesting. Given that CLIP is such a significant way of training vision language models, the inverse scaling law may yield high impact in this direction as well.

- The writing is overall good except some minor points (see in the next section). The visualization in Figure 2 and 3 well demonstrate the shortening strategies used, making it easier to understand for readers who can only afford a quick scan. The results presented in Figure 4, 5, amd 6 are clear, and easy to follow.

- The experimental results are strong. As shown in Table 3, one can train the same arch without much drop (e.g., 2-4% drop) but saving 10x-20x GPU hours. This is can be very important for academic labs where resources are not huge.

- The figure 7 shows that, when given a fixed number of computation budget, you can achieve better accuracy with larger models with this inverse scaling laws applied.

**Weaknesses:**

I did not see major flaws of this paper, but with some minor comments:

- firstly, I think this paper misses a subsection or paragraph in Related works, which can talk about similar findings in language models. To my best knowledge, it has been well known that, larger models can be trained with less steps/epochs/durations compared with smaller models. One example is the paper "Scaling Laws for Neural Language Models". So I think it would be nice to give readers a full spectrum of knowledge by including a series of such related work in the large language domains, and I would not think including them will make this paper less innovative, rather this paper becomes more complete.

- this paper claims multiple times that CLIP is the first foundation model that connects images and text. I am less comfortable about this claim, since there are a bunch of image-text relevant work before CLIP, e.g. such as image captioning model. Having not trained with modern large models does not mean that they are not foundational or they could not have been foundational models if they were trained with large models. This is a minor point, but would love to see more rigorous and nice claim.

- Why there is performance improvement for block mask in Figure 4, and similarly why this is happening fro Figure 5 too. And it seems there is in consistency between different tasks in terms of which strategy can improve or not?

- Why it improves a lot for ImageNet-A in Figure 6, while not the case for other datasets/tasks?

- in the paragraph of line 204, it says syntax masking even enhances the performance for CLIP. I wonder if this is mainly because it keeps nouns, which are more informative for tasks relevant to object recognition. What about some other tasks that focuses on predicting/modeling relationships between objects? Would syntax masking still effective, or would still text masking useful at all?

**Questions:**

see above

**Limitations:**

I would not say this is the limitation, but it pushes me to wonder how general this inverse scaling law is? e.g., would it be easily applied to other tasks?

---

> ### Author Rebuttal · Authors · 2023-08-09
>
> We first thank the reviewer for the detailed comments and the appreciation of our work. We address the concerns below
>
> ### Q1: Missed related works of scaling law in language models
> Thanks for this suggestion! We completely agree that discussing these related works will make this paper more complete, as we share a similar message — that using larger models doesn't necessarily entail a proportionate increase in training computation. We will fix this in the next version.
>
> ### Q2: Rephrase the statement of CLIP
> Thanks for this suggestion. We will add a detailed discussion about the prior efforts on image-text learning and clearly posit CLIP as one of the pioneering works that push this research direction at scale.
>
> ### Q3 & Q4: Performance improvement with masking & Significant improvement on ImageNet-A
> Intuitively, learning with more tokens comes with more information and generally should lead to better performance. But regarding the interesting observation in Figures 4 & 5 & 6 that learning with fewer tokens sometimes leads to improved performance, one possible conjecture is that masking at a low ratio might act as a form of regularization — by filtering out potentially noisy or redundant information, it could help the model to focus more on key features, thereby enhancing representation learning. We believe a more rigorous and systematic investigation of this phenomenon would offer valuable insights and leave it as a future work.
>
> Overall, these findings, while intriguing in their own right, well support the core message of this paper — that the larger the image/text encoders used, the shorter the sequence length of image/text tokens that can be applied in training.
>
> ### Q5: Text masking in objects relationship modeling tasks
> Good question! Our intuition agrees that nouns are particularly relevant or useful for object recognition tasks, therefore the learned representation may not be directly effective for tasks that require a more complex understanding of relationships between objects. However, it is essential to note that our training pipeline contains two stages — the reduced-length input is only applied during pre-training to accelerate CLIP training, and full-length input is used during fine-tuning to ensure that the models access all available information in the training data. This two-stage process enables nuanced modeling of relationships between objects. We will add this discussion in the next version.
>
> ### Q6: Generalizability of inverse scaling law
> We are optimistic about the potential for this inverse scaling law to extend to other image-text learning tasks. This optimism stems from two main observations: 1) Image and text data frequently include redundant information, and 2) Larger models typically possess enhanced representation learning capabilities. We are excited to keep exploring this direction.

---

> > ### Comment · Reviewer_Mhr1 · 2023-08-13
> > **Thank you for the response**
> >
> > Thank you authors for the rebuttal.
> >
> > One idea about the Q5. Maybe it's not too hard to try the "Attribution, Relation, and Order (ARO) benchmark", given their colab has been released for evaluation.
> >
> > I have read other reviewers' comments, I agree that the current evaluation, i.e., zero-shot, is limited, though VITA is included in the rebuttal. After a more comprehensive understanding, I think the current score is a proper reflection of how I view this work, so no change.

---

> > > ### Author Response · Authors · 2023-08-20
> > >
> > > Thank you for these additional valuable suggestions, which are important for furthering the quality of our work.
> > >
> > > ### Q1: ARO Benchmark
> > > We genuinely appreciate your suggestion regarding the ARO benchmark [1]. Our preliminary results are shown in the table below. We can observe that, while OpenCLIP B/16 slightly outperforms CLIPA B/16, the absolute performance of both on this benchmark remains somewhat limited. This aligns with the observations from [1], emphasizing that **CLIP models generally exhibit limitations in capturing relationships, attributes, and order information**.
> > >
> > > To mitigate this relational understanding issue, a composition-aware hard negative mining strategy (NegCLIP) is introduced in [1]. Note that this strategy is extremely lightweight, and can be seamlessly integrated as an additional fine-tuning stage in enhancing CLIP’s text understanding ability. Our results below also corroborate the efficacy of NegCLIP, e.g., both OpenCLIP and CLIPA nearly double their performance on benchmarks like COCO-Order and Flickr30k-Order.
> > >
> > > We will add the discussion above in the next version. We hope that these additions, coupled with the insights from [1], can help alleviate reviewers’ concerns about the capabilities of CLIP’s text encoders.
> > >
> > >
> > > |     Model     | NegCLIP |  VG-Relation | VG-Attribute |  COCO-Order  | Flickr30k-Order |
> > > |:-------------:|:----------------:|:------------:|:------------:|:------------:|:---------------:|
> > > | OpenCLIP-B/16 |                  |     44.7     |     59.9     |     41.8     |       45.3      |
> > > | OpenCLIP-B/16 |     &#x2713;     | 78.6 (+33.9) |  69.5 (+9.6) | 87.6 (+45.8) |   89.1 (+43.8)  |
> > > |  CLIPA-B/16   |                  |     43.8     |     57.1     |     37.8     |       39.1      |
> > > |   CLIPA-B/16  |     &#x2713;     | 78.5 (+34.7) | 68.0 (+10.9) | 86.1 (+48.3) |   87.9 (+48.8)  |
> > >
> > >
> > >
> > > ### Q2: Zero-shot Evaluation
> > >
> > > We recognize and understand the concerns regarding the evaluation breadth. But it is essential to emphasize that our primary focus on zero-shot performance is not arbitrary. One of the most distinguishing facets of CLIP, setting it apart from other learning paradigms, is its strong capability in zero-shot recognition. This unique strength is consistently underscored in both the official CLIP repository and the widely-recognized OpenCLIP repository. Our research approach aligns with this established practice, and extensively validates the inverse scaling law by measuring models’ zero-shot abilities across various datasets.
> > >
> > > While our current study offers insights into one (important) aspect of CLIP's capabilities, we do acknowledge that understanding in a broader context, especially in different downstream tasks, is also crucial. We are interested in and committed to keep exploring these insights in our future endeavors.
> > >
> > >
> > >
> > > **References**
> > >
> > > [1] Yuksekgonul M, Bianchi F, Kalluri P, et al. When and Why Vision-Language Models Behave like Bags-Of-Words, and What to Do About It?[C]//The Eleventh International Conference on Learning Representations. 2022.

---

> > > > ### Comment · Reviewer_Mhr1 · 2023-08-21
> > > >
> > > > Thank you for your efforts on the ARO Benchmark benchmark. It makes sense to see that CLIPA underperforms CLIP, because of language drop out. I encourage the authors to include these results in the final version.

---

> > > > > ### Author Response · Authors · 2023-08-21
> > > > >
> > > > > Yes, these discussions will be included in the final version.
> > > > >
> > > > > Thanks again for your invaluable suggestion about the ARO benchmark and NegCLIP training algorithm.

---

### Author Rebuttal · Authors · 2023-08-09

### Global Response:

We would like to first thank all the reviewers for their invaluable comments and recognition of the interest and value in our discovery of the inverse scaling law in CLIP training (Reviewer Depv, TeGF, Mhr1), enabling the potential to improve accessibility to large models and significantly reduce training costs (Reviewer UF9M, KzVG). We also value the reviewers’ positive feedback regarding the clear presentation of our paper and the extensive experiments conducted.

In the rebuttal below to each reviewer, we carefully address all the concerns raised. Additionally, we have provided a rebuttal file with figures and tables to enhance comprehension, specifically illustrating the inverse scaling law in relation to text tokens, model size, total training tokens against performance, and presenting the new evaluation results on VTAB benchmark.

Lastly, we are excited to share our latest scaling results, benchmarking our model against the current state-of-the-art. By training on DataComp-1B [1], our CLIPA-H/14 delivers an impressive performance of 81.8%, establishing itself as the best ImageNet-1k zero-shot model among released, open-source weights to date. Importantly, this milestone was reached with a training cost under $15,000 (measured using cloud A100 GPU price). This additional scaling result further confirms the efficiency and effectiveness of our proposed CLIPA.

**References**

[1] Gadre, Samir Yitzhak, et al. "DataComp: In search of the next generation of multimodal datasets." arXiv preprint arXiv:2304.14108 (2023).

---

### Decision · Program_Chairs · 2023-09-21

**Decision:**

Accept (poster)

**Comment:**

This paper studies the impact of token length (image and text) on the quality of image-language pretraining in CLIP-style models. The main finding and contribution of the paper is in CLIP-style training, the larger the model is, the shorter the token sequence is required to maintain a certain level of performance. Based on this finding, this paper proposes to train larger models with shorter sequences, which is more computationally favorable than training smaller models with sequences of full length.

This leads to 69.3% top-1 zero shot image accuracy with just eight A100 GPUs in ~4 days, which is 15x speed up over the conventional training protocols.

The paper shows many strong contributions as pointed out by multiple reviewers. 1) the work is well motivated. Reducing the cost of pretraining foundation models has a broad impact. 2) Given that CLIP is one of the founding breakthroughs of vision language models, the lessons identified in the paper may yield high impact. 3)  The writing is overall good. 4) Work provides a comprehensive analysis of various token reduction strategies for both image and text and also proposes new ones. 5) Analysis of different scales of models, across different masking strategies, is really helpful to convert a previously under-explored finding into a consistent phenomenon that can be useful for future research. 6) The work makes large model training accessible demonstrated by strong experimental results. Authors show that one can train the same architecture without much drop (e.g., 2-4% drop) but saving 10x-20x GPU hours.

Few weaknesses that is not fully addressed after the rebuttal is:
- The paper does not address the concern that the limited evaluation of the proposed model may hide model deficiencies compared to the original CLIP.
- Concerns on hyperparameter choice difference between proposed method and baselines
- Novelty vs "RECLIP"; (pointed out that this is concurrent so worth mentioning as a concurrent work with comparison)

To elaborate, one of the main concerns raised from Reviewer `KzVG` is whether what the paper proposes is deemed to be called "inverse scaling law". Neither it is "quantitative" showing mathematical functional trend nor in conventional neural scaling law sense, as a function of optimal compute budget, size of model is increasing. "scaling law" is often reserved for the predictive formula which the paper lacks at the moment.

While the term "scaling law" has not been very concretely defined in the field, the current form the author is using in the title and the paper is slightly misleading and confusing. AC strongly encourages this to be changed; i.e drop "law" (maybe replaced by "relationship" ) and be precise what is scaling with each other ( model size vs sequence length). The author used the term "inverse relationship between the model size and the length of image/text tokens in CLIP training" is more descriptive than "inverse scaling law".

While the findings in the paper are interesting and practical, phrasing in terms of "inverse scaling laws" is not clearly representing the contribution and potentially misleading for people who are familiar with neural scaling laws.

Another issue raised by `KzVG` and `UF9M` is whether evaluation is not sufficient to truly say reduced context training is neck-to-neck with the original method. This is quite important to characterize since the models trained with less compute budget may be missing something the full model learned. As CLIP-style model is used as pre-training to obtain interesting language-vision aligned representation, there could be strong downstream application impact.

Final recommendations were split among the reviewers, three reviewers recommending acceptance (strong accept 8, accept 7, weak accept 6) and two reviewers recommending rejection (weak reject 4, reject 3).  While the concerns raised by two reviewers are valid, the strength of contribution and strong results outweigh potential concerns. As suggested above, AC strongly encourages the authors to rephase "inverse scaling law" as well as make suggested changes from the reviewers. On the topic of insufficient evaluation, AC believes study of better evaluation of representation learned through efficient training would make the paper strong and methodology more impactful but should not hinder publication.